

# Control-Oriented Modelling of Wind Direction Variability

Scott Dallas[1], Adam Stock[2], and Edward Hart[3]

[1]Wind and Marine Energy Systems and Structures, Department of Electrical and Electronic Engineering, University of Strathclyde, Glasgow, United Kingdom
[2]Institute of Mechanical Processes and Energy Engineering, School of Engineering and Physical Sciences, Heriot-Watt University, Edinburgh, United Kingdom
[3]Wind Energy and Control Centre, Department of Electrical and Electronic Engineering, University of Strathclyde, Glasgow, United Kingdom

**Correspondence:** Scott Dallas (scott.dallas@strath.ac.uk)

**Abstract.** Wind direction variability significantly affects the performance and life-time of wind turbines and wind farms. Accurately modelling wind direction variability and understanding the effects of yaw misalignment are critical towards designing better wind turbine yaw and wind farm flow controllers. This review focuses on control-oriented modelling of wind direction variability, which is an approach that aims to capture the dynamics of wind direction variability for improving controller per-

formance over a complete set of farm flow scenarios, performing iterative controller development, and/or achieving real-time closed-loop model-based feedback control. The review covers various modelling techniques, including large eddy simulations (LES), data-driven empirical models, and machine learning models, as well as different approaches to data collection and pre-processing. The review also discusses the different challenges in modelling wind direction variability, such as data quality and availability, model uncertainty, and the trade-off between accuracy and computational cost. The review concludes with a

discussion of the critical challenges which need to be overcome in control-oriented modelling of wind direction variability, including the use of both high and low-fidelity models.

## 1  Introduction

Present day large scale wind farms contain arrays of ever increasing numbers of multi-megawatt turbines, with total capacities in the order of gigawatts. The largest wind farm project in the world, under construction in Gansu Provence China, will contain

around 7,000 turbines and is planned to have a capacity of 20 GW over an approximate area of 500 km squared. The continued increase in the size of wind farms as well as in the size of wind turbines themselves, has resulted in greater interactions between turbines and their surrounding flow fields. These interactions are driven by both large scale atmospheric effects, such as topographically generated weather systems, and more local effects, such as those due to terrain and the wakes of other turbines (Meyers et al., 2022). These complex interactions within the wind farm result in high levels of wind farm performance

uncertainty that can lead to under-performance, threatening the viability of wind power to meet the expectations of future renewable energy targets (Haupt et al., 2017).

Active yaw control (yawing the turbine rotor to face against the incoming wind) and wind farm flow control (using control systems to reduce wake effects on downstream turbines) has motivated research into wind field variability by the wind energy





community. General wind field variability is present in Gaussian wind fields simulated via the turbulence models recommended
by EC 61400-1, the Mann and Kaimal models (Yassin et al., 2021). Direction variation in these models is seen through
the argument of the resultant velocity vector of the lateral and longitudinal components. Although useful for fatigue load
calculations, research has tended to focus solely on the high frequency wind field content approximated by these models at
turbine locations (Dong et al., 2021). Therefore, there is limited understanding of the physical and statistical nature of wind
direction variation on length and time scales important for yaw and wind farm flow control (in the order of metres to kilometres
and seconds to minutes). Furthermore, the behaviour of wind turbines and wind farms under realistic wind direction variation
remains understudied (Shapiro et al., 2022).

This review presents the current understanding of wind direction variability in the context of control-oriented modelling of
wind turbines and wind farms in a manner suitable to a wide audience. In doing so, essential gaps in the literature are highlighted
and areas in need of further research are made clear. The review is motivated partly by the fact that significant unintentional
yaw misalignment (yaw error) of horizontal axis wind turbines (HAWT) with respect to the inflow direction, of more than $10°$,
is common in many wind farms (Annoni et al., 2019a). This amount of persistent misalignment would result in a conservative
decrease in annual energy production (AEP) of more than $\approx 3\%$ (Pedersen et al., 2008), alongside a corresponding increase
in wind energy levelised cost of energy (LCOE). AEP aside, there are also the implications of asymmetric loading through
turbine components, which could cause increased operation and maintenance costs, further increasing the LCOE (Bartl et al.,
2018). Research is ongoing as to the full extent of yaw misalignment on turbine performance and a lack of consensus prevails
in the literature, however there are obvious performance implications.

The review begins in Section 2, where the physical drivers of wind field variability at different length and time scales are
presented and discussed. Section 3 then outlines the various physical and statistical models used to understand wind direction
variability across wind farms over the length and time scales relevant for yaw and wind farm flow control. Next, Section 4
gives an overview of the performance implications of yaw misalignment, both in terms of power and loads. The review then
moves on to the topic of control, starting with Section 5 which details conventional yaw controllers and their associated errors
and uncertainties. This is followed by Section 6, where methods that augment the control system to improve sensor quality
and reliability including methods which utilise machine learning are described. Section 7 then explores two wind farm control
methods affected by wind direction variability, namely wake steering control and collective yaw control. Finally, in Section
8, the critical challenges of control-oriented modelling of wind direction variability are summarised and, in Section 9, the
conclusions are drawn.

## 2 Physics of Direction Variability

Early research towards understanding dynamic wind direction behaviour began in the field of atmospheric science. Researchers
were focused on understanding and predicting the dispersion of pollutants in the atmosphere (Davies and Thomson, 1999). It
was found that wind direction variability came in either the form of gradual meandering of the wind vector (Kristensen et al.,
1981; Hanna, 1983; Etling, 1990) or frequent sudden changes in direction (Mahrt, 2008). The behaviour was also found to be





very closely related to concurrent meteorological and physical conditions, such as the ambient wind speed, the atmospheric stability, local topography, pressure and turbulent motion (Kau et al., 1982). In the wind energy community, wind direction is often treated as as a categorical variable (Simley et al., 2020), or as a conditional variable for direction-dependent coefficient estimation (Feijóo and Villanueva, 2017). In reality, wind direction is a continuous variable with a strong auto-correlation structure (Vincent, 2010), where slight changes can have significant affects on wind farm performance (Porté-Agel et al., 2013). Understanding how wind direction varies over the relevant length and time scales for yaw and wind farm flow control is therefore essential to quantifying performance and achieving control objectives.

Firstly, Section 2.1 gives an overview of the physical processes which cause general inflow variability at wind farms, as well as providing a brief introduction to important terminology from atmospheric science. Next, in Section 2.2, some of the relevant processes in the study of wind direction variability are highlighted and the modelling of these processes is further explored.

## 2.1 Physical Processes

Wind farms experience an array of weather phenomena, resulting in fluctuations in the wind field at different spatial and temporal scales. A subset of meteorological processes and where they fall on the length and time scale is shown in Figure 1.

The largest scale, the synoptic scale, covers atmospheric changes at horizontal length scales in the order of 1,000 km and above, and time scales of approximately one month. The dominant influence on the development of phenomena at the synoptic scale arises from the Coriolis acceleration affecting the movement of air masses (Coleman and Law, 2015). Synoptic-scale processes are mostly relevant for long-term wind energy resource assessment studies (Spera, 1994).

The next largest scale is the mesoscale. Mesoscale meteorology is the study of atmospheric phenomena characterised by horizontal scales in the order of 1 km to 1000 km. Time scales at this level cover less than a day to several weeks. The phenomena often of most interest encompass thunderstorms, fronts, and topography/terrain driven weather systems such as mountain waves (Coleman and Law, 2015). Mesoscale processes influence the location choice and long term operation of wind farms as well as driving smaller scale processes which can affect wind farm performance directly (Spera, 1994).

Lastly, the microscale encapsulates atmospheric phenomena on the smallest scales. These phenomena generally occur over time scales of seconds to minutes and length scales in the order of 1 km or less. This scale focuses on individual thunderstorms, clouds, and local turbulence arising from structures like buildings and obstacles such as individual hills (Coleman and Law, 2015). Microscale processes affect the everyday operating environment of wind farms, as they lead to turbine inflow variability on short time scales which can negatively impact performance. These negative impacts are what yaw or wind farm flow control systems are designed to mitigate (Haupt et al., 2015).

Many of the microscale processes that occur are so transient in nature that the deterministic description and forecasting of each individual deviation from the general flow of the fluid (eddy) is almost impossible. As a result, there are three primary areas of research regarding the characterisation of eddies (Stull, 1988), which are

– **Stochastic methods** which deal with the empirical average statistical properties of the eddies, these are often studied through simulations using Reynolds-averaged Navier–Stokes (RANS) equations (Section 3).





**Figure 1.** Scales of atmospheric motion and example phenomena.

– **Similarity theory** which describes the apparent common-behaviour of many empirically-observed phenomena, when transformed to the relevant scale. Similarity theory has been applied to wind farm flow data to determine inputs to RANS equations (Breedt et al., 2018).

     – **Phenomenological classifications** which inform a partially deterministic approach towards the larger sized eddy structures, these are often studied through large eddy simulation (LES) (Section 3).

**2.2    Wind Direction Variability**

The variability of the wind direction depends highly on the inverse of the wind speed and the stability conditions of the atmospheric boundary layer (ABL). The ABL is the lowest part of the Earth's atmosphere, which directly interacts with the Earth's surface. The height of the ABL depends on various factors such as weather conditions and the time of day. The





behaviour of the ABL is often described through it's stability condition, which is categorised by three main cases: highly
turbulent (unstable), nearly laminar and intermittent (stable), or a combination of the two (neutral) (Meyers et al., 2022).

On both the microscale and mesoscale, different sets of dynamics can dominate depending on the ABL conditions, which
can have significant effects on wind direction variability and wind farm performance (Meyers et al., 2022). In an unstable
atmosphere, small scale convective processes are of most importance in determining the variability of the wind direction
(Vincent et al., 2010). This variability is well understood through ABL similarity theory of turbulence (Hans and Jhon, 1984).
On the contrary, in a stable atmosphere, larger scale processes are able to exist, such as inertial oscillations, low-level jets,
gravity waves and Kelvin-Helmholtz instability, which tend to dominate the variability (Stull, 1988).

Application of traditional similarity theory under stable conditions predicts a reduction in direction variations as stability
increases. However, as a consequence of low frequency meanders (Hanna, 1983), this was shown to fail for averaging times of
more than 10 minutes (Davies and Thomson, 1999). Low frequency meandering has been attributed to boundary-layer motion
and larger mesoscale effects (Hanna, 1983). Low-frequency meanders have been found to exist over all types of terrain and
even over the open ocean, and various formulas for estimating such effects have been proposed (Hanna, 1983, 1990; Joffre and
Laurila, 1988).

In addition to slow meandering motion, the wind is known to abruptly change direction as well. The underlying mechanics of
sudden local wind direction changes remain poorly understood, but potential factors include steepening gravity waves, density
currents, pulses of drainage flow, and numerous other more complex phenomena that are difficult to model and predict (Mahrt,
2011).

Another crucial aspect is the correlation between wind direction variability and the inverse of wind speed (Joffre and Laurila,
1988; Davies and Thomson, 1999). On average, wind direction variability tends to be higher for unstable conditions at a
given wind speed. However, very low wind speeds occur more commonly in stable conditions. As a result, the wind direction
variability is generally much larger at night because of the relatively shallow and stable nocturnal boundary layer (Mahrt, 2011).
During the late night is also when wind veer (the clockwise rotation of wind direction with height) is especially pronounced as
a result of Coriolis forces on the nocturnal boundary layer (Porté-Agel et al., 2020).

The inverse relationship between wind direction variability and wind speed has been successfully modelled and generalised
(Joffre and Laurila, 1988; Hanna, 1990). The models help account for wind direction variability with increasing height and
between different atmospheric stability classes (Mahrt, 2011). These generalised models have limited application in wind farm
flow modelling, since they tend to focus on regimes with very low wind speeds (and therefore high wind direction variability)
when most turbines would not be operating.

Finally, terrain effects are also know to impact wind direction variability. In mesoscale simulations, direction variability was
found to be greater in complex terrain over small averaging times (in the order of minutes) and showed high sensitivity to the
grid point selected to represent the on-ground conditions (Jiménez and Dudhia, 2013). Nevertheless, this distinction becomes
indiscernible for averaging periods of more than 10 minutes, suggesting that local complex terrain predominantly induces
greater wind direction variability on shorter time scales, but still significant for yaw and wind farm flow control (Mahrt, 2011).





## 2.3 Discussion of Physics of Direction Variability

The overall drivers of wind direction variability at wind farms is a combination of large scale effects at the synoptic or mesoscale
as well as local effects at the microscale (Vincent, 2010). Over longer time periods of several hours (in both stable and unstable
conditions) synoptic and mesoscale eddies are the main contributors to wind direction variability (Davies and Thomson, 1999).
Certain variation occurs regularly and follows predictable patterns, such as that arising from diurnal and seasonal cycles. Other
variations are more sporadic, driven by large-scale weather systems that can induce abrupt changes in wind speed and direction
(Haupt et al., 2019). On the other hand, at the microscale, aspects such as atmospheric stability, terrain effects and wake effects
are the main drivers of variability.

Each of the drivers of wind direction variability exist on different length and time scales meaning that the statistical properties
of wind direction measurements constantly change. Even on very long time scales, climate change ensures that there is no
time scale on which the measurements can definitely be considered stationary, meaning that the associated data has means,
variances, and co-variances that constantly change over time (Vincent, 2010). Non-stationarity makes it difficult to use physical
phenomena as indicators to inform and adjust the parameters of the control system, however, atmospheric stability dependent
readjustment time of yaw control parameters has been tested (Cortina et al., 2017).

Fundamentally, there may not be one single direction associated with the wind flowing into large wind farms, especially
for those surrounded by complex terrain (Quick et al., 2020). The challenge therefore is to understand how wind direction
measurements need to be first filtered and conditioned, before optimisation for control objectives can occur (Hau, 2013). The
degree of filtering and conditioning needed will in general depend on other factors such as the concurrent wind speed and
atmospheric stability, alongside other site-specific factors like topography, terrain and the specifications of the yaw system
itself.

## 3 Wind Farm Flow Models

Wind farm flow models are mathematical, statistical and/or computational models used to simulate and analyse the behaviour
of wind flow within wind farms. Many different flow models exist that take into account various different global and/or local
effects, however, they have traditionally been developed by various research communities in isolation (Sanz Rodrigo et al.,
2017). Recently, attempts have been made to bridge the gaps, especially between the fields of atmospheric physics, statistics
and fluid dynamics, where collaboration is motivated by the need for realistic inflow conditions in high fidelity wind farm flow
studies (Chatterjee et al., 2018).

One question is whether or not a sufficient picture of the relevant physics can be captured by wind farm flow models,
such that they can be used for controller testing and validation in a reliable and accurate way. Current developments in high
fidelity farm flow models would allow for such simulations to be performed, although at great computational expense. Thus,
the number of processors required to achieve useful results using these high fidelity models is out of reach to the majority of
the researcher community.





Section 3.1 discusses the current physical models that have incorporated realistic dynamic wind direction changes as input and briefly describes how they work. Section 3.2 then follows with discussion of the statistical tools and models that have been applied to the study of wind direction variability over the relevant length and time scales.

## 3.1 Physical Models

Physical models used in wind farm flow simulations fall into one of three broad categories; high fidelity large eddy simulations
(LES), medium fidelity dynamic models or reduced order (engineering) models.

- **High fidelity LES models** are the most accurate but still computationally feasible microscale farm flow simulation tools available. Instead of direct numerical simulations of the Navier-Stokes equations of fluid dynamics, LES works by filtering out the smallest length-scales of the Navier-Stokes equations (the smallest eddies). Generally, LES is used to simulate statistically stationary behaviour of wind farms, however, realistic dynamic wind direction variation can be
included by coupling LES with mesoscale forcings that prescribe the wind farm inflow through precursor simulation methods (Section 3.1.2) (Munters et al., 2016).

- **Medium fidelity dynamical models** can be employed to predict the available power and/or flow fields in a wind farm (Boersma, 2019). These equations often use Reynold's averaged Navier-Stokes (RANS) equations based models, which, unlike LES, represent only the mean fluid flow. RANS models mostly consider steady-state behaviour, but models can
be adapted to analyse preset changes in wind farm conditions over space and time, such as continuous sweeps across inflow directions (Kheirabadi and Nagamune, 2021).

- **Reduced-order or engineering models** can provide information on important wind farm dynamics with limited computational complexity (Boersma, 2019) which give typical run-times in the order of seconds to minutes, useful for iterative controller design. However, these models are valid for only specific atmospheric conditions, don't contain any true tur-
bulent eddy structure and have limited accuracy (Schreiber et al., 2020).

      LES models are the highest fidelity models available and have been used successfully for testing new wind farm flow controllers (Storey et al., 2016; Gebraad et al., 2016). The quality of these models are constantly being improved by validation against and assimilation of field test data, as well as recent attempts to couple them with mesoscale precursor models (Munters et al., 2016; Chatterjee et al., 2018; Stieren et al., 2021). However, the grid points needed to resolve a developed stratified
wake with LES is in the order of $1 \times 10^{11}$, according to conservative estimates (Li et al., 2022). Hence, the computational cost is prohibitively expensive for most controller design purposes, not to mention the cost associated with the wind turbine aero-elastic models required to gain a complete picture (Larsen et al., 2017). Therefore, LES is not suitable for most control-oriented modelling applications. Instead, LES often serves as a proof-of-concept tool for new control methods or as validation models for lower-order surrogate models (Meyers et al., 2022).
The best available models for understanding the effects of wind direction variability are coupled mesoscale-microscale LES models. Although, again these models are unsuitable for most control-oriented modelling applications, they are able to





simulate farm wide realistic dynamic changes in inflow direction and have provided valuable insights. Therefore, Sections 3.1.1 and 3.1.2 describe in more detail mesoscale, microscale and coupled models and introduce examples from the literature.

### 3.1.1 Mesoscale and Microscale Models

Mesoscale models of wind farms include physical parameterisations to model the outer flow phenomena by including energy transform models, surface layer models, land use models, physical parameterisation, boundary layer parameterisations, and more. By incorporating suitable initial and boundary conditions derived from global models, these models effectively capture the dynamic processes of the ABL (Haupt et al., 2019). These important dynamics are often excluded from or only roughly approximated in more local LES (microscale) models. Furthermore, mesoscale models are non-hydrostatic and model water-

related processes in the atmosphere, both rare features of microscale models. Although realistic wind direction variability can be captured using mesoscale models (Draxl et al., 2021), the maximum horizontal resolution of these models is too large to allow them to accurately investigate intra-wind farm effects caused by dynamic wind direction changes (Carvalho et al., 2012; Jiménez and Dudhia, 2013).

In contrast to mesoscale models, microscale LES models, have the ability to capture the flow around objects at much higher

resolution, allowing modelling of terrain details and flow around turbine blades (Haupt et al., 2020). These models are also able to resolve fine-scale turbulence and explicitly resolve aeroelastic interactions with the wind turbines. Microscale LES models, therefore, are essential towards developing new optimal yaw and wind farm flow control strategies (Fleming et al., 2014a, 2015). However, up to now the emphasis has been on small-scale turbulence modelling and scenarios where the farm flow is constrained towards steady-state conditions (Calaf et al., 2010; Wu and Porté-Agel, 2011; Goit et al., 2016). While

these simulations have offered valuable insights into the interaction of wind farms and the ABL under steady-state conditions, the influence of large-scale effects on wind farm performance, especially dynamic wind direction changes, has mostly been ignored (Stieren et al., 2021).

### 3.1.2 Coupled Models

There have been efforts to accurately couple mesoscale models to microscale LES (Muñoz-Esparza et al., 2014; Muñoz-Esparza

and Kosović, 2018; Haupt et al., 2020), which is particularly important to accurately represent non-stationary meteorological conditions or changes of atmospheric stability at wind farms, especially those driven by the diurnal cycle (Haupt et al., 2020). For coupled simulations, Coriolis effects are included which means large changes in wind direction with height in the ABL can be simulated (Haupt et al., 2017). Therefore, in order to represent a wider range of important meteorological phenomena that affect wind farm performance, mesoscale information needs to be embedded in microscale models (Draxl et al., 2021).

Realistic inflow conditions from mesoscale forcing can be included in microscale LES by nesting the LES within a mesoscale numerical weather prediction (NWP) simulation domain. The output of the NWP acts as a precursor to the LES simulation, providing both the initial and boundary conditions. Examples include coupling LES to mesoscale models like the Weather Research and Forecasting (WRF) model (Talbot et al., 2012; Mirocha et al., 2014; Schalkwijk et al., 2015). Biases in wind speed and direction in nested mesoscale simulations have been shown to be passed on to the LES simulations, which in general





are unable to fully correct for these biases (Talbot et al., 2012). However, the wind field is reasonably well simulated by the WRF model, especially in wind regimes where there is a very dominant sector (Carvalho et al., 2012), and can be improved with appropriate data assimilation techniques (Haupt et al., 2017).

     The goal of accounting for realistic dynamic wind direction or even sweeps over a range of predetermined wind directions in LES is challenging and demands significant computational resource. To this end, a concurrent precursor method in which

the horizontally periodic mesoscale precursor domain was rotated was first proposed by Munters et al. (2016). Following up on this work, Chatterjee et al. (2018) proposed a modified version of the concurrent method that only rotated the inflow velocity vector instead of the entire precursor domain. Data from cup and vane anemometer was used to generate realistic neutral ABL inflow data to the modified model to compare the predicted wake effects with on-site light detection and ranging (LiDAR) measurements of the wakes (Chatterjee et al., 2018). The approach has since been developed further by Stieren et al. (2021)

to make use of a dynamically changing non-inertial rotating reference frame, which was able to accurately reproduce realistic pseudo-random wind direction and power spectrum at each turbine using low-pass filtered wind farm field measurements as inputs.

     The coupled LES models provide greater understanding of how dynamic wind direction changes can significantly impact wind farm performance. As an example, simulations of a regularly spaced wind farm array demonstrated a considerably steeper

decline in power output at the minimum farm power inflow angle, $\theta_{min}$ (the wind direction at which lowest wind farm power output occurs), during a dynamic wind direction sweep compared to what was predicted through a series of static simulations at various but constant inflow directions (Munters et al., 2016; Stieren et al., 2021). The drop in power was explained by the high-velocity wind speed channels which exist between turbines. The flow in these channels was much stronger during static simulations at $\theta_{min}$ compared to simulations which considered a sweep over directions, where channel flow is disrupted by

the inflow angle, especially between turbines further downstream (Stieren et al., 2021). The effect was less pronounced for low wind direction rotation rates, since the channel flow had enough time to speed up and allow the entrainment of energy from the channels into the waked region (Munters et al., 2016). This effect also produced a spike in wind farm power at wind farm flow angles far away from $\theta_{min}$. It also was shown to cause a site specific hysteresis effect, detected as a positive or negative shift in the value of $\theta_{min}$ of the wind farm (Munters et al., 2016; Stieren et al., 2021).

## 3.2   Statistical Models

Statistical models are useful as inputs to wind farm simulations in order to account for and accurately reflect uncertainty in the inflow conditions. Since wind direction is fundamentally non-stationary, this necessitates simplifying assumptions and approximations about the statistical nature of wind direction time series so they can be more easily modelled. In general, there is a relative lack of research focusing on the statistics of wind direction as opposed to wind speed (Jiménez and Dudhia,

2013), especially in the context of wind farm flow, which seems to be a product of the challenges associated with the statistical treatment of circular variables like wind direction (Mardia et al., 2000). Often in studies, the longitudinal and latitudinal components of the wind vector are shown instead of the wind direction, which avoids the difficulties associated with summary statistics of circular data (Haupt et al., 2017).





Therefore, one critical question is how to treat the circular wind direction variable. In contrast to linear statistics, there are often different ways to calculate summary statistics of circular data, such as the sample mean for instance, which in most cases give different results. Therefore, careful consideration of the appropriate circular statistics is needed, before making any calculations (Farrugia and Micallef, 2017).

### 3.2.1 Circular Statistics

Circular statistics deal with data that has a circular or directional nature, where the values need to be measured in terms of a circular scale. In contrast to traditional linear statistics, where values can be measured with respect to the real number line, circular statistics takes into account the wrapping of the variable, where any value beyond the maximum or minimum are wrapped back on the scale, creating distributions that exist on the circle rather than the real number line. Wind direction provides a good example of a circular variable. It is $2\pi$ periodic and can be mapped to a circular scale where an arbitrary zero-direction and manner of rotation are defined (Jammalamadaka and SenGupta, 2001). Conventionally, the zero-direction is set as north and then angles are measured clockwise from there.

The periodicity of circular variables, the arbitrariness of the zero position and manner of rotation, and the absence of absolute magnitude, altogether means directional analysis of circular data is substantially different from standard linear statistical analysis. Circular statistical methods need to be invariant with respect to the choice of the zero-direction and sense of rotation, as a consequence, many typical linear techniques and measures are not applicable. Therefore commonly used summary statistics, such as the sample mean and variance, as well as simple mathematical operations like subtraction and addition, need to be redefined so they make sense in the context of circular statistics (Jammalamadaka and SenGupta, 2001).

Before the circular mean and variance can be discussed further, the absolute minimum angular distance $|\Delta(\theta_1, \theta_2)|$ needs to be defined. This quantity gives the absolute value of the least angular distance between two angles (represented by $\theta_1$ and $\theta_2$ here), and is defined as,

$$|\Delta(\theta_1, \theta_2)| = \min(|\theta_2 - \theta_1| \bmod 2\pi, 2\pi - |\theta_2 - \theta_1| \bmod 2\pi). \tag{1}$$

The associated sign of the minimum angular distance $\Delta(\theta_1, \theta_2)$ depends on the choice of the zero-direction and the sense of rotation (Farrugia et al., 2009).

The minimum angular distance is needed to compute the expected values of essential summary statistics, like the circular mean and standard deviation (Yamartino, 1984; Farrugia et al., 2009). The circular mean $\bar{\theta}$ and circular variance $\sigma_\theta^2$ of a sample of $N$ circular variables $\{\theta_i\}_{i=1}^N$ can be obtained in a variety of ways. The most intuitive is the vectorial method and involves representing the circular data as a set of unit vectors in the complex plane $\{z_i\}_{i=1}^N$. The circular mean is then calculated as the argument of the resultant vector,

$$\bar{\theta} = \arg\left(\frac{1}{N}\sum_{i=1}^N z_i\right). \tag{2}$$





Figure 2 illustrates calculation of the circular mean of two different sets of circular variables. Note that for wind data, calculation of the circular mean can also be weighted by the corresponding wind speed $V_i$ in order to capture more information about the wind field.

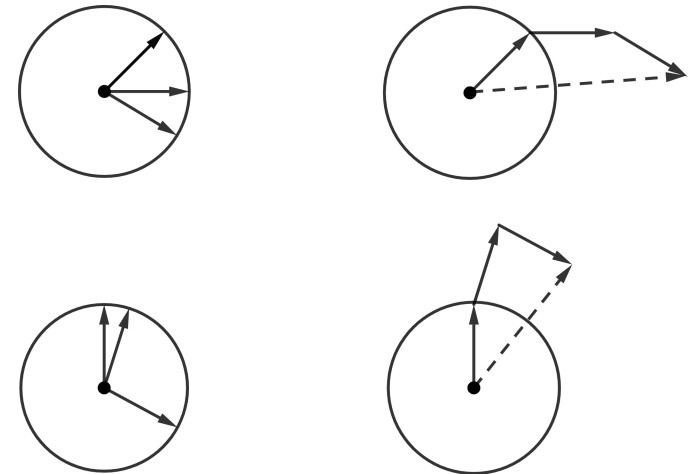

**Figure 2.** Examples of circular variables represented as vectors and resultant vectors indicated by dashed lines. The circular mean is then the argument of the resultant vector in each case. Adapted from Cremers and Klugkist (2018).

Once the circular mean $\bar{\theta}$ is obtained, the circular variance $\sigma_\theta^2$ can then be calculated according to,

$$\sigma_\theta^2 = \frac{1}{N} \sum_i \Delta_i^2 - \left( \frac{1}{N} \sum_i \Delta_i \right)^2 , \tag{3}$$

where $\Delta_i = \Delta(\theta_i, \bar{\theta})$. This quantity is of particular interest since the standard deviation of wind direction $\sigma_\theta$ is related to the lateral turbulence intensity $i_v$ through the equation $\tan(\sigma_\theta) = i_v$ in stable atmospheric conditions (Hanna, 1983).

### 3.2.2 Short Term Statistical Models

In order to quantify variability for robust wake steering control, where upstream turbines operate with an intentional yaw misalignment to deflect their wakes away from those downstream (Simley et al., 2021), statistics of 5 minute wind direction time series have been studied from one second wind vane met mast data (Rott et al., 2018). The measurement data was split into 5 minute time series, mapped to a linear scale, and compared with a fitted normal distribution both visually, using histograms and quantile-quantile plots, and numerically, using a Kolmogorov–Smirnov test. The comparison was done to verify the hypothesis that the measurement data can be approximated statistically by a normal distribution within 5 minute segments. It was found that 70.58% of the measurements passed the test for a significance level of 5%. Based on these findings, it was concluded that in the majority of cases, the variability of 5 minute wind direction time series can be adequately approximated by a normal distribution. It was also verified that wind direction variability is strongly correlated with atmospheric stability classes, as discussed in Section 2, which included stable, neutral and unstable conditions (Rott et al., 2018).



Similarly, it has also been shown that a normal distribution provided a good fit to the measured wind direction variations over a longer 10 minute time period at Horns Rev (Gaumond et al., 2014). The wind direction measurements were recorded using a sonic anemometer mounted on a met mast with a sampling rate of 12 Hz at a height of 50 meters (Peña and Hahmann, 2012). The assumption that the wind direction time series was normally distributed over the considered sampling times meant that the yaw misalignment errors at each turbine could be assumed to be normally distributed as well, which allowed power performance to be more accurately calculated. Hence, the accuracy of three separate wake models was evaluated against data from the Horns Rev wind farm while taking into account uncertainty in the wind direction measurements.

Alternative data driven methods for modelling and generating realistic short term wind field time series samples have also been described (Bossanyi, 2018; Simley et al., 2020; Van Der Hoek et al., 2021). Bossanyi (2018) started from single-point measured data, which were 10-minute averages of wind speed, direction and standard deviation from a met mast. To preserve the correct 10-minute statistics, smooth time-series were fitted to the points and synthetic turbulence was then added. While the wind field included all three components of turbulence, the lateral component was zero-mean, therefore dynamic changes in inflow wind direction were subsequently added from the smoothed met mast data.

Alternatively, both Simley et al. (2020) and Van Der Hoek et al. (2021) modelled the wind direction by generating different stochastic time series which represented either the slowly varying mean wind direction across the wind farm or the purely turbulent high frequency component with zero mean. The time series were produced by simulating a random time series with a normal distribution, derived from the power spectra of both low-frequency and turbulent wind direction components extracted from met mast measurements and LES. This method resulted in time series where the low frequency wind direction components were completely correlated at each turbine whereas the high frequency components were completed uncorrelated.

Strong assumptions are made by these data driven models, especially in how wind direction changes propagate through the farm, however, data-driven methods are designed to minimise computation requirements and act only as a starting point to be iterated and refined upon. Other, more general wind field generation techniques are also available and widely used, such as the Mann spectral model (Mann, 1998) or the Veers method (Veers, 1988), however, these methods focus mostly on modelling stationary processes and the high frequency content of the wind field.

### 3.3 Discussion of Wind Farm Flow Models

Meso-microscale coupled LES models have the potential to validate a controllers effectiveness under realistic wind direction variability before more detailed field tests are carried out (Section 3.1.2). However, the computing power required by current models makes them prohibitively expensive and time consuming to deploy, especially for complex control optimisation (Munters et al., 2016; Stieren et al., 2021).

Ideally, software would allow many multiple 5 to 10 minute wind farm flow simulations to test controller effectiveness under dynamic wind changes, enough to achieve statistical significance. Although current data-driven methods make strong assumptions about wind direction, especially in terms of normality of time series and their spatial and temporal coherence, the short-term statistical treatment of the wind direction variable presented in Section 3.2 provides a starting point for a data-driven, computationally less expensive approach to the problem.



# 4 Performance under Yaw Misalignment

Yaw misalignment refers to any misalignment between the rotor axis with the hub height wind direction. Figure 3 shows the top down view of a turbine with positive yaw misalignment.

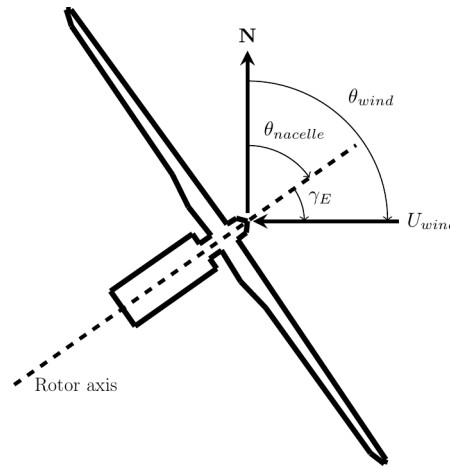

**Figure 3.** Positive yaw misalignment $\gamma_E = \theta_{wind} - \theta_{nacelle}$ on a horizontal axis wind turbine which is defined as a counter clockwise rotation of the rotor viewed from above. Adapted from Fleming et al. (2016).

There are two classes of yaw misalignment; intentional, because of the actions of a wake steering controller (or simply because of the necessarily slow actuation of the yaw system), or unintentional, because of systematic measurement bias or other errors in the wind turbine measurement equipment.

This section starts by providing motivation for the topic through the physical laws that govern horizontal axis wind turbines. Section 4.1 covers the first-order relationship between power and yaw misalignment of the wind turbine. Then, Section 4.2 gives a brief overview of the current understanding of the effects of yaw misalignment on turbine loads.

## 4.1 Power

From the continuity equation of fluid mechanics, the flow of an air mass $\dfrac{\mathrm{d}m}{\mathrm{d}t}$ is a function of air density $\rho$, surface area (in this case the rotor swept area) $A_r$, and free stream flow velocity $U_\infty$. Ignoring the effects of wind sheer and veer, it is estimated that the velocity is independent of location on the rotor swept area, meaning that $\dfrac{\mathrm{d}m}{\mathrm{d}t}$ through the rotor can be defined as,

$$\frac{\mathrm{d}m}{\mathrm{d}t} = \rho A_r U_\infty. \tag{4}$$

The instantaneous kinetic power of the wind available at the rotor, $P_w$, is

$$P_w = \frac{1}{2}\frac{\mathrm{d}m}{\mathrm{d}t}U_{wind}^2 = \frac{1}{2}\rho A_r U_\infty^3. \tag{5}$$



A wind turbine exerts a thrust force $F$ on the wind flowing through the rotor that corresponds to the amount of energy extracted from the flow each second,

$$F = \frac{1}{2} C_T(\beta, \lambda, \gamma) \rho A_r U_{wind}^2, \tag{6}$$

where $U_{wind}$ is the free-stream wind velocity after taking into account induction effects and $C_T(\beta, \lambda, \gamma)$ is the dimensionless thrust force coefficient, which is a function of the blade pitch $\beta$, tip-speed ratio $\lambda$, and yaw angle $\gamma$. The tip-speed ratio is defined as the ratio of the tangential speed at the blade tip to free-stream wind velocity,

$$\lambda = \frac{\omega R}{U_\infty}, \tag{7}$$

where $R$ is the radius and $\omega$ is the rotational speed. The tip-speed ratio is proportional to the rotor speed, which is typically
controlled via the generator torque or by pitching the turbine blades to alter the lift forces on them (Boersma et al., 2017). The power in the wind across a circular cross section was given in eq. 5 but not all of this power can be extracted by a wind turbine. The wind power that can be extracted by a turbine is given by,

$$P = \frac{1}{2} C_p(\beta, \lambda, \gamma) \rho A_r U_{wind}^3, \tag{8}$$

where $C_P(\beta, \lambda, \gamma)$ is the dimensionless power coefficient (Boersma et al., 2017).

The theoretically maximum available power at any given wind speed occurs when the rotor axis is aligned to the inflow wind direction. If the rotor axis of a turbine is not aligned with the inflow, the wind speed perpendicular to the rotor plane is reduced to

$$U_{\gamma_E} = U_{wind} \cos(\gamma_E), \tag{9}$$

where $\gamma_E$ is the yaw misalignment of the turbine. Hence, neglecting changes in aerodynamic behaviour from misaligned rotors,
the maximum amount of power that can be extracted by a turbine operating with a yaw error $\gamma_E$ is

$$P_{max} = \frac{1}{2} \rho A_r U_{wind}^3 \cos^3(\gamma_E) C_p. \tag{10}$$

Thus, the extractable power is theoretically reduced by a factor of $\cos^3(\gamma_E)$. In reality, experimental results have shown that power extraction under yaw error behaves according to the more general empirical equation,

$$P_{max} = \frac{1}{2} \rho A_r U_{wind}^3 \cos^\alpha(\gamma_E) C_p, \tag{11}$$

where the term $\cos^\alpha(\gamma_E)$ is referred to as the power reduction factor (PRF). The $\alpha$ term has been estimated both experimentally and theoretically in several different studies, which are discussed in Section 4.1.1.

### 4.1.1  Power Reduction Factor

Experiments carried out using a rotating wind turbine model in a wind tunnel with turbulent inflow generated by a static grid found that the empirical value of the power reduction factor mostly agreed with the expected value, i.e. $\alpha \approx 3$ (Krogstad and



Adaramola, 2012). A similar set up with low and high turbulence uniform inflow and sheared inflow condition also found that $\alpha \approx 3$ (Bartl et al., 2018). However, other experimental results have often shown that the cube law overestimates the power loss (Kragh and Hansen, 2015). An overview of past research and their findings is shown in Table 1.

| Turbine Model | $\alpha$ Value | Paper |
|---|---|---|
| Scale model | $\approx 3$ | (Krogstad and Adaramola, 2012; Bartl et al., 2018) |
| Scale model | $\approx 2$ | (Medici, 2005) |
| NREL 5MW LES model | 1.88 | (Gebraad et al., 2014) |
| Scale Model | $\approx 1.7870$ | (Schreiber et al., 2017) |
| Scale model, LES model | $\approx 1.43, \approx 1.43$ | (Draper et al., 2018) |
| Envision 4MW turbine, LES model | $\approx 1.86, \approx 1.73$ | (Fleming et al., 2017) |
| Various OEM models | $\approx 2$ | (Howland et al., 2020) |

**Table 1.** Selected details of past research and findings for the power reduction factor $\cos^{\alpha}(\gamma_E)$.

In addition to the empirical observations outlined in Table 1, Howland et al. (2020) developed a model from first principles, using blade element momentum (BEM) theory to show how there exists a non-linear relationship between power output and

yaw misalignment, affected by both the atmospheric conditions and the wind turbine control system. The data collected to test their model showed $\alpha = 2$ for different OEM turbines at a specific site. It was concluded that the ability of the first principles model to accurately predict performance was much greater than the simple cosine cubed power law, since the expected power will in all cases be model- and site-specific.

### 4.2 Loads

Fatigue loading occurs when a load is repeatedly applied and removed from a material, i.e. when the loading is cyclic. For wind turbines, cyclic loads usually occur as the blade rotates through a wind field, leading to what is called once-per-revolution (1P) loads on the blade and 3P loads on the tower and drivetrain (Kragh and Fleming, 2012). The effects of yaw misalignment on turbine component and structural fatigue loads as well as lifespan changes is somewhat of an open question (Bartl et al., 2018).

A misaligned inflow produces periodic loads because the aerodynamics of the blade change with its azimuthal position $\theta$.

The advancing and retreating action of the blade with respect to the crosswind flow creates a change in the angle of attack, leading to changes in the lift, drag and thrust forces (Heck et al., 2023). The changes in thrust force combine to create a moment on the rotor in the tilt direction. Figure 4 shows a free body diagram of a blade element before and after applying a positive yaw misalignment. As the blade passes through $\theta = 0$ and $\theta = \pi$, the effect of the misalignment is at a minimum since it is cancelled by the blade position, whereas the effect is maximal at $\theta = \pi/2$ and $\theta = 3\pi/2$. Additional periodic loading occurs

because of a slow down in the turbine's wake on one side compared to the other, which results in increased forces on the blade during that portion of the rotation (Zalkind and Pao, 2016).

Damage equivalent load (DEL) is the single equivalent load at some fixed frequency that produces the same amount of damage as the actual loading history. The distribution of DELs and extreme loads under yaw misalignment for various degrees





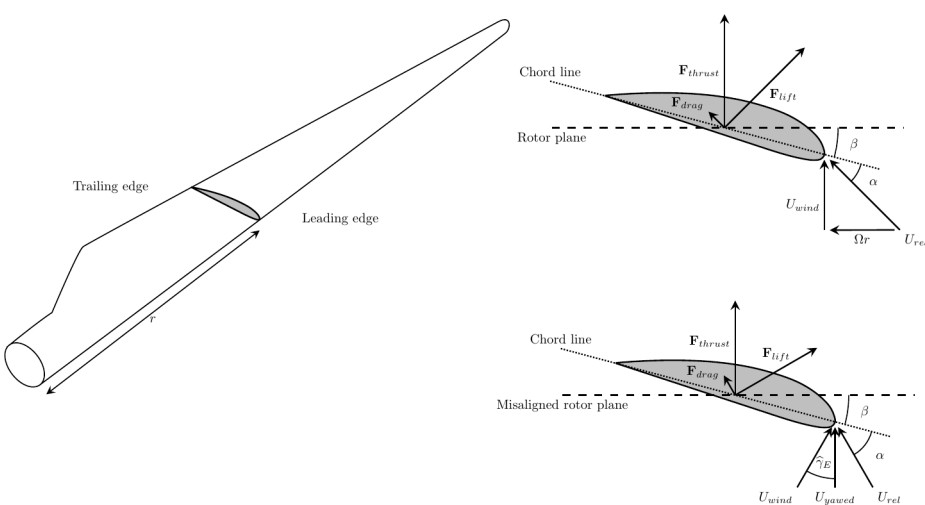

**Figure 4.** Blade element dynamics under normal and yawed conditions, $\widehat{\gamma}_E = \gamma_E \sin(\theta)$ and $U_{yawed} = U_{wind} \cos(\widehat{\gamma}_E)$. Adapted from Howland et al. (2020).

of yaw misalignment have been found to be rather complex but correlated with the rotor and blade design as well as the ambient wind conditions (Damiani et al., 2018). These load distributions were measured for a fully instrumented wind turbine and compared to predictions from an aeroelastic model, where it was found that the model predicted the distributions well (Damiani et al., 2018). Modelling deficiencies in other aeroelastic models and complex unsteady-flow phenomena during yaw were also revealed by comparison of load characteristics on a misaligned model turbine rotor to various computational approaches (Schepers et al., 2014).

More recently, it was shown how the DELs are not distributed symmetrically around the zero misalignment angle on the turbine's main bearings (Cardaun et al., 2019). In fact, it was found that rotating the rotor to the right of the inflow lead to smaller loads in general. This effect has since been attributed to the rotor tilt, which, at $\gamma_E = 0$, results in a minor increase in the effective wind speed on one side of the rotor while reducing it slightly on the other side (Hart et al., 2022). Similarly, the yaw moments on misaligned rotors were observed to increase approximately linearly with increasing degrees of yaw misalignment but again the moments were not completely symmetrically distributed around the zero misalignment angle (Bartl et al., 2018).

It has been argued that the effects of yaw misalignment can be balanced by wind shear, such that there exists a turbulence-intensity dependent optimal non-zero yaw misalignment angle which minimises blade loads (Kragh and Hansen, 2014; Damiani et al., 2018). However, the reduction in blade loads at this angle were shown to be accompanied by an increase in load fluctuations for other components, such as the drivetrain and tower (Kragh and Hansen, 2014; Zalkind and Pao, 2016).

### 4.3 Discussion of Performance under Yaw misalignment

The performance effects due to misalignment between the rotor and the inflow wind direction are complex and dependent on a number of factors including the turbine model and the ambient wind conditions.





Levels of yaw misalignment greater than 10 degrees are not an uncommon occurrence according to the literature (Pedersen et al., 2008, 2011; Kragh and Fleming, 2012; Annoni et al., 2019a). Figure 5 highlights typical mean and maximal misalignment
angles as well as power losses expected at different values of power reduction factor. From Figure 5, it can be seen that commonly found levels of yaw misalignment in the literature can cause anywhere from an $\approx 1.5\%$ to $\approx 4.5\%$ decrease in AEP.

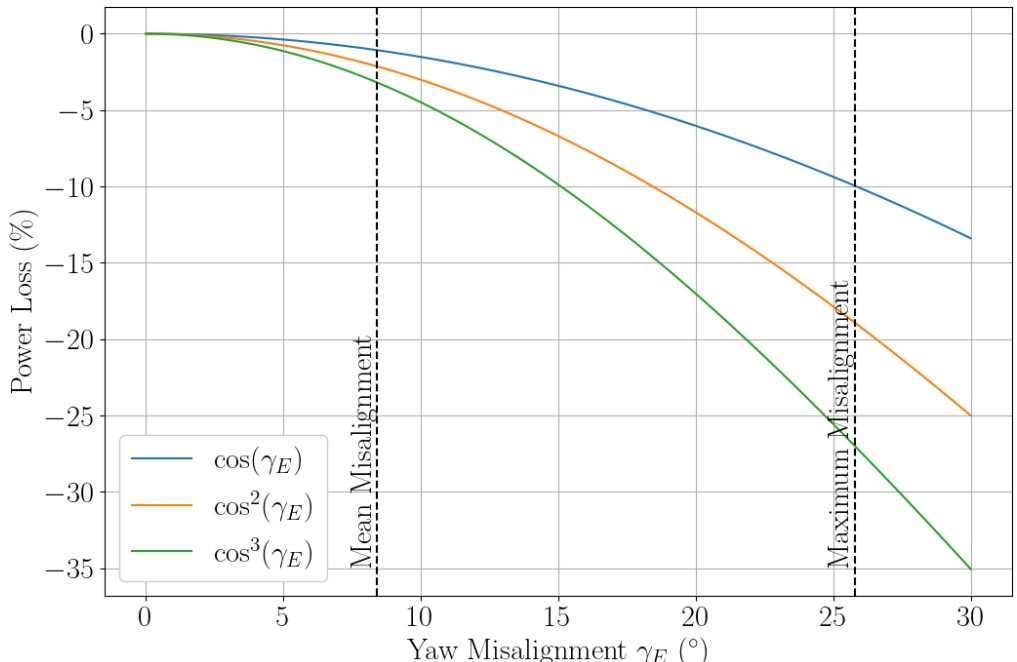

**Figure 5.** Power loss against misalignment for different power reduction factors with typical mean and maximal yaw misalignment values indicated from data presented by Annoni et al. (2019a).

Yaw misalignment also causes asymmetric loading through the blades and rotor, leading to increased wear and tear on the components of the turbine, reducing their lifespan and increasing maintenance costs, with knock on effects on LCOE
(Section 4.2). Although the blade loads under yaw misalignment have been well described and verified in multiple studies, more understanding of the aerodynamics of yaw misalignment is still required, including differences between positive and negative misalignment angles, as well as how the rotor is affected by both vertical and horizontal variations in direction (Howland et al., 2020).

The first order approximation of yaw misaligned rotor dynamics (Figure 4) provide a good starting point in understanding
site and atmosphere specific effects of yaw misalignment on power and loads (Howland et al., 2020). Then, if these dynamics are integrated into aero-elastic turbine simulations, control-oriented models could be developed with these dynamics in place, resulting in better understanding of the efficacy of control actions to minimise the deleterious effects of yaw misalignment.





## 5   Conventional Yaw Control

The rotational movement of the wind turbine rotor around the axis of the turbine tower is the yaw of the turbine (Kragh et al.,
2013b). Yaw controllers are designed to align the wind turbine rotor axis with the hub height wind direction as best as possible,
while balancing the constraints of the system (Meyers et al., 2022). As discussed in Section 4, the wind turbine's yaw system
can have significant effects on overall wind turbine performance in terms of both power and loads.

It is important to note that the control architecture of commercial wind turbines is often proprietary and dependent on the
manufacturer, and so information on the operation of conventional wind turbine yaw systems is only available to a limited
extent in the literature. The discussions in this section, therefore, may not be true for all wind turbines but they do serve as
motivation for further discussions on alternatives to conventional yaw systems.

This section begins by describing the architecture of conventional yaw control systems in Section 5.1. Then, the common
errors and uncertainties associated with conventional wind direction measurement instruments are discussed in Sections 5.2
and 5.3 respectively.

### 5.1   Architecture

The majority of modern utility-scale horizontal axis wind turbines use an active yaw drive mechanism to face the turbine into
the wind. An estimate of current wind direction is the first step in most yaw systems. Traditionally, a wind vane on top of
the nacelle measures the wind direction at a point behind the rotor plane. The wind direction signal is usually measured at
high frequency by the wind vane (Bossanyi, 2019). The wind direction signal is then passed through a heavy low-pass filter,
which smooths out the short-term variations, makes the resulting signal more representative of rotor-averaged variations and
ensures the yaw system depends only on the relatively low-frequency changes in the wind direction. As an example, a first-
order low-pass filter with a -3 dB cut-off frequency of 2 mHz was applied to the input wind direction in CFD simulations of
yaw control (Gebraad et al., 2016). The filtered signal is then compared with the nacelle orientation to obtain a measure of yaw
misalignment. An example of typical conventional yaw control architecture is shown in figure 6 (Chen et al., 2020).

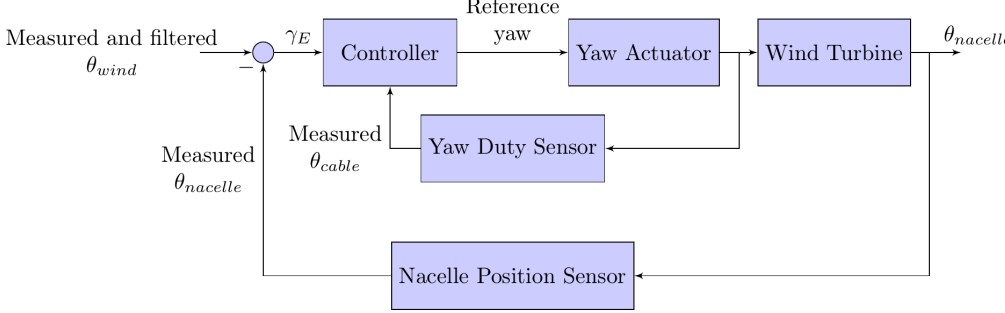

**Figure 6.** Schematic of a typical conventional yaw system. The yaw duty sensor measures cable rotation $\theta_{cable}$ to ensure the rotation remains
within safe limits. Adapted from Chen et al. (2020).





In addition to low-pass filtering, a hysteresis dead-band, effectively a buffer zone where no control action is taken, is introduced to prevent frequent yaw manoeuvres and avoid dangerous gyroscopic forces. This avoids what is known as 'yaw hunting', where the yaw controller tries to follow the time-varying wind direction too closely without allowing for an amount of variability and uncertainty in the signal. If the turbine were to yaw at such a high rate, this would have negative consequences on the lifetime of the yaw system as well as the loads on other components. Most large turbines yaw at rates of less than 1

deg/s (Pao and Johnson, 2009) and the controller is typically only activated when the yaw error measured by the wind vane exceeds some threshold (Spencer et al., 2013). An example from the literature comes from the baseline controller from the pre-design phase of the DOWEC 6MW turbine (Kooijman et al., 2003). This controller used a 30 second moving average of the wind direction to monitor yaw misalignment. The controller activated yaw actuators when the yaw error reached 5 degrees with a yaw rate of 0.3 degrees per second until the 2 second moving average of yaw error was less than 0.5 degrees (Storey

et al., 2016).

Due to the constraints described, the yaw system is in standstill most of the time (Kim and Dalhoff, 2014). It is typical for the yaw angle to remain constant for about 5 to 10 minutes before the yaw control corrects for the changes in wind direction and reduces the yaw misalignment (Rott et al., 2018). The contrast between the slowly reacting yaw systems of modern utility-scale wind turbines and the variability of the wind direction signal is a product of the trade off between minimising yaw duty and

yaw hunting while at the same instance maximising turbine performance.

## 5.2    Measurement Errors

Wind vanes or sonic anemometers positioned atop the nacelle within the disturbed flow region behind the rotor are often used to measure the apparent hub height wind direction (Kragh et al., 2013a). On-site met masts are sometimes also available to provide measurements, however it is more convenient in general to use measurements from instruments on the turbines

themselves. Each turbine control system can provide a hub height wind direction estimate by comparing the measured nacelle position against the input low-pass filtered yaw misalignment signal, which all relies heavily on correct calibration of the instrumentation (Bossanyi and Ruisi, 2021).

Measurements taken by sensors positioned behind the rotor on the nacelle, within the disturbed flow, have been shown to be significantly affected by flow distortions caused by the rotor (Kragh and Fleming, 2012). Computational fluid dynamics

(CFD) simulations of the flow distortions around the nacelle revealed a strong sensitivity of the wind direction measurement to the position of the sensor on the nacelle (Zahle and Sørensen, 2011). It was revealed that the nacelle flow angles exhibited substantial variations with height above the nacelle surface. The CFD simulations showed that the flow was primarily governed by unsteady vortex shedding from the cylindrical part of the blades connected with the rotor hub interacting with the root vortices from each of the blades, resulting in the creation of significant velocity gradients. The effect of flow distortion has

also been shown in field studies. Nacelle mounted sensors showed significant dependence of flow distortion on both yaw and tilt angles with yaw error of up to 10 degrees when operating in a tilted inflow (Zahle and Sørensen, 2011). Additionally, analysis of operational data from a V80 2 MW onshore turbine revealed below-rated mean yaw errors of 10° (Pedersen et al.,



2008, 2011), whereas separate analysis of the CART3 600 kW research turbine showed rotor speed–dependent mean yaw errors of $5°$ to $15°$ (Kragh et al., 2013a).

Further inaccuracies can be introduced purely from the way the yaw control system is set up and operated. Firstly, for the Horns Rev I wind farm, analysis of operational data showed the yaw signals to be mostly wrong (Draxl, 2012). Upon restart, with the turbine yawed at a random angle, it took time for the sensor to be oriented correctly again, resulting in a period of inaccurate data (Draxl, 2012). Secondly, complications common to many wind turbines were introduced by the turbines' own cables, which had to be disentangled after too much rotation around the yaw axis, meaning the turbine had to be rotated

back and then re-adjusted against the other sensors again (Draxl, 2012). Lastly, an EU project (UpWind) found that the wind vane signals of both onshore and offshore turbines were often not correctly calibrated, with neighbouring turbines measuring substantial differences in yaw alignment (Eecen et al., 2011).

Biases in turbine wind direction signal can be corrected once they have been identified. For example, a speed-dependent linear regression correction scheme, based on empirical data, was applied to a yaw controller input signal (Kragh et al., 2013a).

With the correction applied, the new yaw control architecture was able to reduce yaw errors compared to the baseline controller. However, the relatively short amount of data available meant the findings could not be properly substantiated, and precluded any additional conclusions about load reductions.

### 5.3 Measurement Uncertainty

An important issue highlighted, especially in wake steering research, is the wind direction uncertainty present in data sets

(Gaumond et al., 2014; Rott et al., 2018; Simley et al., 2020; Campagnolo et al., 2020). This uncertainty is guaranteed due to the stochastic behaviour of the wind. The uncertainty can also be exaggerated through standard methods of time averaging as well as from spatial interpolation, as a result of the natural variability of the wind direction and the distance from the reference location to where the measurement is taken.

Operational data sets are often binned by wind direction sectors in order to simplify the calculation of other important

variables, mainly power production. However, the accuracy of wind farm flow models was found to heavily depend on the width of the wind direction sectors used for binning the simulation results (Gaumond et al., 2012). Hence, over narrow wind direction sectors, differences between the power outputs predicted by wind farm flow simulations and real wind farm power output data sets are potentially caused by the large wind direction uncertainty in the data sets, and not because of modelling deficiencies (Gaumond et al., 2012). As a result, there is now a recognition of the need to incorporate uncertainty into wind

farm flow models to produce better and more robust controllers.

In order to quantify uncertainty in wake models and to design better wake steering controllers, the distribution of high-frequency wind direction measurements within 5-minute (Rott et al., 2018) or 10-minute (Gaumond et al., 2014) windows was approximated using a Gaussian probability density function. By quantifying uncertainty, deficiencies in wake modelling were identified and inflow specific adaptations to wake steering controllers were explored.

Similar approaches inspired by the Gaussian distribution approximation of the wind direction have also been developed. For example, the yaw position uncertainty was included in wake steering set-point calculations alongside the wind direction





uncertainty as a joint Gaussian distribution where the sums of the variance of each equalled the variance of the yaw error (Simley et al., 2020). Another approach used polynomial chaos expansion to account for uncertainties while optimising for wake steering set-points which included a Laplace distribution for the yaw misalignment and a Gaussian distribution for the wind direction measurement (Quick et al., 2020). The polynomial expansion approach revealed that uncertainty in the wind direction measurement had one of the largest impacts on the set-point optimisation results, highlighting the importance of understanding wind direction variability for both yaw and wake steering control.

### 5.4 Discussion of Conventional Yaw control

Control based on conventional sensing methods mainly suffers from two factors. The first is the significant noise, uncertainty and outliers in the inputted wind direction measurement. These problems have been found to be due to a mixture of the placement of the sensing equipment, the inadequacies of standard measurement instruments and the intrinsic complexity of the wind direction variable (Kragh and Fleming, 2012; Kragh et al., 2013a). Secondly, the slow actuation of the yaw system, although necessary to avoid negative gyroscopic forces, results in turbines operating misaligned most of the time (Mikkelsen et al., 2010). The misalignment can be significant, especially when a wind direction change happens rapidly and abruptly before the yaw system has time to respond.

Control parameters of conventional systems are often determined through a trial and error approach (Bossanyi, 2019), which in many cases is sub-optimal and prone to the proliferation of bias (Mikkelsen et al., 2010) (Section 5.2). In most cases, biases can be identified and corrected using simple detection and correction algorithms (Kragh et al., 2013a). The uncertainties, however, are less easily handled, especially those arising from natural variation in the wind direction. One proposed solution is to use an optimisation under uncertainty methodology for robust control, which entails the incorporation of the uncertainties into the calculation of control parameters and set points (Section 5.3).

### 6 Alternative Yaw Control

Research to improve yaw control has focused on alternative sensing or data-processing methods that provide more accurate inputs to the control system and/or provide a preview of wind direction changes before they occur at the turbine. Alternatives can be broadly categorised by how their input signal is obtained; measurement-free, inferred, forecasted, based on improved measurement equipment, or estimated. It is important to note that some of these methods can be complimentary to each other. For instance, estimation techniques can be used to further enhance control based on remote sensing. The categories are described as follows,

- **Measurement-free yaw control** originates from early wind turbine research limited by the measurement and control technology of the time. The mechanism of this set of controllers is to directly search for the maximum power point without a wind direction input signal (Farret et al., 2001; Xin et al., 2012; Karakasis et al., 2016).



– **Inferred signal based yaw control** is where measurements of other closely related variables are used to infer the wind direction and yaw misalignment angle. For example, an estimation of the yaw misalignment in the below rated domain can be calculated from an inverted function of wind power and wind speed (Tsioumas et al., 2017), or from the rotor angular speed (Karami et al., 2021), and then incorporated into the control system with the appropriate architecture. Nacelle-mounted anemometer wind speed measurements are less affected than the wind vane by flow distortions caused by the rotor and are easier to correct for than wind direction measurements at the same location (Smith et al., 2002). Therefore, the measurement errors and uncertainties associated with wind vane measurements discussed in Section 5 can mostly be avoided without the need for additional sensing equipment.

– **Forecasting for yaw control** is where very short term predictions of the time series are calculated in order to preemptively react to any changes in wind direction (Section 6.1).

– **Yaw control with additional or alternative sensing** which could replace or augment nacelle mounted wind vanes. The most popular alternatives are remote sensors based on LiDAR and hypersonic (SoDAR) technologies (Barthelmie et al., 2016) (Section 6.2).

– **Enhanced signal estimation for yaw control** which involves families of both parametric and non-parametric methods of communication based spatial filtering, bias correction and/or error detection. Some of these methods work by updating the parameters of physics-based models to obtain farm-wide direction estimates, whereas others are purely stats based (Section 6.3).

Since the latter three methods directly address the handling of the wind direction signal (forecasting, improved measurement, and estimation), they are discussed in more detail. Firstly, wind direction forecasting for both yaw control and more general purpose is discussed in Section 6.1. Next, in Section 6.2, improved measurement methods are discussed that reduce uncertainty in the wind direction signal. Finally, in Section 6.3, an outline of wind direction estimation techniques that can improve the quality of wind direction signals without any additional or improved sensing equipment is given.

## 6.1 Wind Direction Forecasting

Since the statistical properties of the wind field evolve with time (Section 2), the forecasting of wind direction is an especially complex task (Hirata et al., 2008). Non-stationarity necessitates the use of non-parametric methods and adaptive spectral analysis to produce accurate short term forecasts. The use of short term wind field forecasts for control purposes is motivated by the preview effect, where information about incoming changes to the flow field can be used to preemptively carry out a desired control action. Theoretically, accurate forecasts could improve turbine yaw performance by reducing the time delay between changes in direction and activation of the yaw system. This is especially attractive in a yaw control setting where response time is limited greatly by the slowness of the yaw actuators.

There are four general categories of methods for forecasting wind direction,





– **Persistence methods** assume that the wind direction at time $t$ is the same as at time $t + \Delta t$. Unsurprisingly, the performance of this method is comparable to physical and parametric methods only for very short term forecasts (Hirata et al., 2008; El-Fouly et al., 2008). This approach is the most naive and is only used as a baseline comparison.

– **Machine learning (ML) and statistical methods** have been used several times to forecast the wind direction variable for wind energy applications. The simplest are regression models (linear or piecewise-linear) (Howland et al., 2022a), Kalman filters (Song et al., 2018) and time series models which include various auto-regressive predictors (Erdem and Shi, 2011; Song et al., 2017). The complex nature of wind direction time-series presents challenges when applying these techniques. Parametric-based forecasters, in particular, tend to be susceptible to bias (Kim, 2003), and although they are easy to implement, most of these methods are linear while wind direction time-series are non-linear in nature (Chitsazan et al., 2019).

– **Numerical weather prediction** (NWP) refers to any physics based approach in meteorological forecasting. NWP models tend to be general-purpose models that can be used for a wide variety of applications including wind direction forecasting. In general, the resolution of NWP models is too coarse to be useful for most wind energy applications, however, one study has demonstrated the performance of an extremely high resolution numerical weather prediction model (Chan and Hon, 2016). A maximum resolution of 200 meters was achieved, but required numerous meteorological instruments and large amounts of processing power, making it poorly suited for yaw or wake steering control-oriented applications.

– **Hybrid methods** make use of mixed models from either statistics or NWP alongside artificial intelligence based methods to improve forecasting. For example, gradient boosting trees ML algorithms were combined with feature engineering techniques to extract the maximum forecasting information from a NWP grid (Andrade and Bessa, 2017). Another example used a circular regression based approach, which was developed alongside a Bayesian averaging method for bias correction of the forecasts obtained by NWP models (Bao et al., 2010).

Methods from machine learning and statistics are the most useful for control purposes since they can be implemented at a local level and in real-time, allowing for adaptive adjustments over extremely short time intervals. Therefore, they are discussed further in Section 6.1.1.

### 6.1.1 Forecasting with Machine Learning and Statistics

Several wind direction forecasting methods based on machine learning for yaw or wake steering control have been investigated, including an auto-regressive integrated moving average (ARIMA) model approach paired with a Kalman filter (KF) (Song et al., 2017). ARIMA models are well-suited for capturing short-term correlations and have been used extensively in a diverse mix of forecasting applications (Fisher and Lee, 1994; Bivona et al., 2011). In general, however, the ARIMA model by itself is unable to adjust its parameters effectively as new time-series information becomes available. To solve the adjustment problem, the ARIMA model was combined with a Kalman filter (KF), which assimilates new data and updates the model's parameters systematically (Su et al., 2014; Song et al., 2018). The ARIMA-KF model was able to predict the one step ahead 10 second





mean wind direction with a mean absolute error (MAE) of $0.92°$ over a 4 hour validation window after assimilating 20 hours of training data. When incorporated into yaw control, the new system was able to recover 1-2% of lost power due to yaw misalignment compared to a baseline conventional controller.

A simple linear regression-based method was also used to forecast the wind direction during periods of mean wind direction transitions to produce inputs to various wake steering controllers (Howland et al., 2022a). The linear regression approach resulted in an MAE of $1.3°$ after a time horizon of 30 minutes during transition periods compared to an MAE of $1.9°$ when the low-pass filtered wind direction signal was used. More complex forecasting methods from machine learning have also been explored, including four different data mining algorithm prediction approaches (Ouyang et al., 2017). Support vector machines, neural networks, random forests and gradient boosted regression trees were each trained and tested on a years worth of wind direction data at 10 minute intervals, transformed into cosine and sine components. Although it was found that the methods based on random forests and neural networks performed best at predicting the 10 minute ahead sine and cosine components of the wind direction, performance improvements by integration of forecasts into the yaw system were not demonstrated.

## 6.2 Improved Sensing Equipment

Various different solutions have been suggested which use advanced sensing equipment to improve the wind direction input signal to the yaw control system. One way is to augment or replace the wind vane with a LiDAR system mounted on the nacelle, on the ground or on the rotating spinner of the turbine to detect the undisturbed wind in front of the turbine over the entire rotor (Mikkelsen et al., 2013; Simley et al., 2014; Fleming et al., 2014b; Scholbrock et al., 2016). By installing a spinner anemometer in front of the rotor, the measurements are likely to be less influenced by rotor-induced flow distortions, offering advantages over measurements obtained from a sensor placed behind the rotor (Kragh et al., 2013a). Simulations demonstrated that a spinner mounted continuous wave LiDAR can estimate yaw misalignment with a median precision below $4°$ (Kragh et al., 2011). In field tests, good correlation was found between estimates of yaw error determined using a spinner mounted LiDAR and those estimated based on met mast data (Kragh et al., 2013b). Further field tests also demonstrated how a nacelle mounted LiDAR can correct measurements from a nacelle mounted wind vane, resulting in increased yaw alignment and significantly improved power capture compared to the uncorrected baseline case (Fleming et al., 2014b).

Similar to forecasting techniques, LiDAR and other remote sensing methods can allow for further performance gains by providing wind field preview information to the yaw control system. A LiDAR capable of providing preview wind direction information for the next 60 seconds, harnessed using conventional model predictive control (MPC) in the yaw system, could yield an 8% increase in power production and potentially lead to reductions in fatigue loads during instances of extreme wind direction changes (Spencer et al., 2013). Likewise, the performance of a yaw control system with access to preview information from forward facing LiDAR coupled with a long-short term memory neural network was tested against a conventional yaw control system in simulations (Chen et al., 2020). It was found that incorporating preview information could increase power capture by up to 3.5%, reduce yaw travel by up to 5.3%, and reduce yaw events by up to 3.9%.

Other advanced measurement technologies similar to LiDAR have also been tested, namely RaDAR and SoDAR. For example, a spinner anemometer consisting of three SoDAR sensors performed well in field tests (Pedersen et al., 2008), although it





is unclear if such devices are commercially available yet. Other improvement techniques involve the use of additional conven-
tional measurement equipment placed strategically around the wind farm in order to better characterise the inflow (Chen et al.,
2022).

## 6.3 Wind Direction Estimation

As discussed in Section 2, the wind direction can vary greatly spatially and temporally due to variable meteorological condi-
tions, local topography and wake effects. Therefore, on top of the possible misalignment biases on local direction measurements
discussed in Section 5.2, the direction is often different at different locations in the wind farm. Hence, in a lot of cases, even in
the presence of enough sensors and/or advanced sensors, it is still difficult, if not impossible, to get an accurate global picture
of wind direction. Under these conditions, distributed wind direction estimation techniques can be considered.

The earliest example explicitly for control purposes was presented by Doekemeijer et al. (2018). A non-linear Kalman filter
was used to assimilate data and update the parameters of a medium-fidelity physical wind farm flow model with the objective of
achieving real time closed-loop wake steering control. However, only high frequency changes in wind direction were accounted
for by the model, such that a constant mean value was assumed over the entire simulation time interval. In order to address
lower frequency changes in wind direction, Sinner et al. (2020) used a simpler polynomial based Kalman filter and updated the
parameters of the model through the assimilation of SCADA data. The major benefit of this approach is the ability to provide
smooth wind direction estimates, even in the case of faulty individual turbine sensors, while only using measurements already
collected at the wind turbines.

Non-parametric methods have also been developed to estimate the wind direction. In the work by Annoni et al. (2019a),
comparisons were made between different non-parametric approaches for estimating the wind direction at turbine locations.
The most accurate of these methods in terms of MAE was a distributed consensus-based optimisation approach. This approach
was shown in simulations to reliably estimate the wind direction across a wind farm even when faults and/or biases were
introduced in the wind vane signals. The MAE of the consensus-based approach was $2.99°$ compared to $3.78°$ for the best
averaging based approach, weighted averaging, and $8.41°$ when using the sensors alone. Additionally, Bossanyi (2019) also
investigated weighted averaging methods for improving wind direction estimates. Short 30 minute wind farm simulations
showed that these methods improved yaw control performance and by extension wind farm power production compared to
using only the turbine's wind vane signal (Bossanyi, 2019).

More recently, Van Der Hoek et al. (2021) applied Gaussian process (GP) regression to the problem of wind direction
estimation. GP regression is a non-parametric Bayesian approach to regression (Rasmussen, 2003), which can be used not
only to estimate the wind direction at any point within the wind farm, but also for bias detection and correction. Thus, the GP
approach provided a balance between the qualities of the parametric and non-parametric methods previously described. Van
Der Hoek et al. (2021) found that a simple GP model with a squared exponential kernel was able to filter the high-frequency
component of artificially generated wind direction data and reproduce the known low-frequency wind direction variation at
turbine locations better than standard low-pass filtering. However, there was no discussion around the choice of kernel to





calculate the covariance or interpretation of model hyper-parameters, both of which needed further exploration to improve the model's accuracy.

It is important to point out that in order to test these estimation techniques, in most cases it was necessary to generate an artificial 'true' wind direction signal as input to the simulations (Section 3.2). This entailed making strong assumptions about the 'true' wind direction, which limits how applicable the results of this section are to real world conditions. Nonetheless, these methods provide an indication of how to best generate realistic and dynamic wind direction changes which could serve as inputs to control-oriented models.

### 6.4 Discussion of Alternative Yaw Control

Errors in measurement of the wind direction at each turbine can be reduced through a variety of alternative and novel methods. The reduction in errors results in overall performance improvements, often without any adaptation or augmentations to the turbines themselves and with minimal alteration to the control architecture.

Forecasting methods, for example, have harnessed the preview effect to preemptively yaw; reducing misalignment errors and improving wake steering controllers (Howland et al., 2022a) (Section 6.1). Similarly, remote sensing equipment such as LiDAR systems have been shown to improve performance through the same effect by measuring the incoming wind some distance in front of the turbine, while also improving wind direction sensing in general (Section 6.2). However, remote sensing technology comes with the added costs of the equipment itself, the expertise needed to operate them effectively and uncertainties in how much turbine performance can be improved by their use (Spencer et al., 2013). Therefore, the relative size and cost of the wind farm needs to be taken into account before making any decisions, since any improvements in performance and reduction in loads may not be substantial enough to justify the extra costs.

Estimation methods such as spatial filtering have been shown in limited simulation scenarios to reduce signal uncertainty and boost overall yaw controller performance without any changes to the actuators or sensing equipment (Bossanyi, 2019; Annoni et al., 2019a; Van Der Hoek et al., 2021) (Section 6.3). Spatial filtering can also make use of the preview effect in downstream turbines by passing information from turbines further upstream (Bossanyi, 2019).

Although these results are all promising, it is fundamentally difficult to rigorously characterise the effectiveness of wind direction forecasters, sensors and estimators, particularly due to the difficulties in generating 'true' wind direction signals to compare them against. Indirect indicators like power production can be used instead, however these will in general be much more sensitive to the wind speed rather than the wind direction, hence caution needs to be taken when setting benchmarks.

### 7 Wind Farm Flow Control

Wind farm control (WFC) considers the entire wind farm as a control system, with individual turbines acting as agents in a network, helping to achieve farm-level objectives (Sinner et al., 2021). Wind farm flow control (WFFC) is a subfield of WFC where the control objective is achieved through manipulation of the intra-wind farm flow. Two promising developments in the area of WFFC are wake steering control, and communication-based spatial filtering, which aims to enhance the accuracy and





reliability of information used by turbine- and farm-level controllers by combining together wind field measurements gathered
from individual turbines (Sinner et al., 2021).

This section briefly introduces examples of both wake steering control, in Section 7.1, and communication-based spatial
filtering for yaw control (designated as collective yaw control), in Section 7.2. The examples represent a small subset of
available control methods but are chosen as they are designed to handle wind direction input variability directly.

## 7.1 Wake Steering Control

Wake steering control provides an example of WFFC sensitive to wind direction changes. Although it can be achieved through
various methods, this section focuses on the the most popular method found in the literature, the use of static yaw misalignment
of upstream turbines. Similar to the objectives of yaw control, in wake steering control, the goal is to balance yawing frequently
enough to maintain power maximisation while avoiding overuse of the yawing components (Houck, 2022). Contrary to the
objectives of yaw control, however, upstream turbines are operated with an intentional yaw misalignment to redirect their wakes
away from downstream turbines, therefore mitigating potentially substantial power losses caused by wake effects (Howland
et al., 2019). Wake steering controllers have been shown to result in farm-wide power performance gains in both simulations
and field experiments (Howland et al., 2022b). Results from one field experiment revealed power production gains of up to
$14\%$ for a downstream turbine over a $10°$ wind direction sector (Fleming et al., 2019), however, the total farm wide power
gains (or in some cases losses) from wake steering control are sensitive to atmospheric conditions, local terrain and the specific
turbine model (Annoni et al., 2018b; Fleming et al., 2019).

Commercial wake steering controllers are available, an example is the WakeAdapt™ software offered by Siemens Gamesa
(Energy, 2022), but the details of their operation is mostly proprietary. Because of this privacy, there is limited information
available on how the software works in general. In the literature, wake steering controllers solve a dynamic optimisation
problem at the wind farm level in order to identify optimal yaw set-points that manipulate the wind field in such a way
that power losses are minimised (Kheirabadi and Nagamune, 2019). These set-points are then tracked by wind turbine level
controllers.

Most wake steering controllers in the literature are designed such that the yaw set-points are optimised under stationary
or steady inflow conditions. This has changed recently by the incorporation of wind field variability into already established
model-based yaw set-point optimisation methods. For example, a steady-state wake model was enhanced by including yaw
system deviations from set-point values in the corresponding wake steering yaw set-point calculations (Quick et al., 2017). This
optimisation approach has since been taken a step further such that the set-point calculations were formulated as optimisation
under dynamic wind direction uncertainty, as opposed to static and deterministic inflow (Rott et al., 2018). Furthermore,
methods for set-point optimisation under uncertainty, with special consideration of wake model parameter uncertainty, resulted
in demonstrable improvements for open-loop and closed-loop wake steering control (Howland, 2021).



### 7.1.1 Graph and Cluster View

A simplification of wind farm flow, particularly in regard to control of the turbines whose wakes interact, is the graph or cluster view of the wind farm. The graph view is an abstraction of the wind farm as a collection of cells, nodes (turbines) and edge weights between nodes which change depending on the incoming wind direction and wake effects. The cluster view similarly groups turbines which are coupled through their wakes. Clusters are defined such that the performance of the turbines in each cluster is only significantly affected by the operation of the other turbines in the same cluster.

Examples of graph-based and cluster-based approaches are those developed by Starke et al. (2021) and by Bernardoni et al. (2022) respectively. The graph-based model proposed by Starke et al. (2021) employed edge weights based on inter-turbine wake interaction intensity and time delays to simulate how the effects of wind direction changes propagate through the wind farm. The graph-based approach employed a Gaussian wake model to calculate velocity deficits and the wake profile (Shapiro et al., 2019). In contrast, the cluster-based approach of Bernardoni et al. (2022) was model-free and used only power data to identify wind direction changes and turbines coupled through wake interactions.

Both types of approaches can lead to efficiency improvements in a distributed control setting and reduce some of the computational challenges associated with real-time control applications, as only the relationships between selected turbines are considered rather than the whole farm or velocity field (Bay et al., 2018; Annoni et al., 2018a, 2019b; Bernardoni et al., 2020). An advantage of the graph-based approach over standard wind farm flow modelling approaches is that it can be integrated with a dynamic wind farm flow model which accounts for changes to wind direction through a time-dependent change in the graph structure. This overcomes the difficulty and computational expense of implementing a dynamic wind change in models that have a fixed domain often with a fixed mean wind direction, such as LES, RANS, or data-driven models trained for a single-inlet condition (Shapiro et al., 2022).

Both the graph and cluster-based approaches provide simplifications for identifying and responding to changes in power output due to changes in wind direction. However, these simplifications are significant and have not been thoroughly validated yet. For example, calculating the correct weightings in the graph-based approach relies on knowing the real wind dynamics, which in turn would ideally need LES or similar to validate. Likewise, the model free cluster-based approach relies solely on power data correlated over time windows in the order of tens of minutes, which introduces limitations on how accurately interacting turbines can be identified and how quickly changes in wind direction are detected. To a greater or lesser extent, both approaches are only able to capture mean wind field effects across the wind farm, which limits their ability to quantify uncertainty in their results as well as for use in a robust control framework.

### 7.2 Collective Yaw Control

Collective yaw control can be achieved through the use of appropriate consensus algorithms for estimating wind conditions at different wind farm locations (Section 6.3). The sharing of data among turbines not only reduces signal noise via spatial filtering (Sinner et al., 2021), it can also help to identify and correct any faults or bias in individual turbine measurements (Annoni et al., 2019a; Van Der Hoek et al., 2021), which not only confers greater control robustness but also extra redundancy





against equipment failures. The reduction in noise and error terms through consensus methods means they can be used to improve yaw and wake steering controller performance through collective yaw control. Table 2 outlines past research and selected findings.

| Software Used | Control Method | Consensus Method | Power Gain | Yaw Duty Reduction | Identifies Yaw Bias | Paper |
|---|---|---|---|---|---|---|
| LongSim | CYC | Weighted average | $\approx 0.2\%$ | $\approx 24\%$ | No | Bossanyi (2019) |
| FLORIS Version 2.1.1 | CYC, CYC + WSC | Weighted average | 0.5%, 4.7% | 46.1%, 17.0% | No | Sinner et al. (2021) |
| Custom in-house | CYC | Gaussian processes | NA | $\approx 20\%$ | Yes | Van Der Hoek et al. (2021) |
| Custom in-house | CYC | Distributed optimisation | NA | NA | Yes | Annoni et al. (2019a) |

**Table 2.** Selected details of past research. CYC = Collective Yaw Control, WSC = Wake Steering Control.


The ability of collective yaw control to improve performance was first demonstrated by Bossanyi (2019) and then by Sinner et al. (2021). The most simple wind direction estimation technique, based on averages weighted by distance from nearby turbines, was investigated in both studies. It was found that power production can be improved over short simulation periods compared to the use of conventional control methods by up to 0.5% in the case of yaw control alone and 4.7% when combined
with wake steering control.

It was also highlighted by Bossanyi (2019) how some turbines in the wind farm can benefit from preview information from the turbines situated further upstream. During 30 minute simulations, the slowly reacting yaw system was able to preemptively activate in anticipation of a change in direction. This effect was found to increase power production, while also reducing both the total yaw travel and the total number of yaw events significantly (yaw duty, Table 2).

An alternative method based on a simple GP regression method introduced in Section 6.3 was investigated by Van Der Hoek et al. (2021). It was found that unnecessary wind turbine yaw activity was reduced by $\approx 20\%$ through the use of an online version of the GP regression method incorporated into a collective yaw control system where the GP model was updated every 10-minutes with new measurements. However, the online model created less accurate predictions over time, indicating more sensitivity to the input data than the offline model and a need for greater refinement of the methodology.

**7.3   Discussion of Wind Farm Flow Control**

The performance of wind turbines clustered together in a farm is inextricably coupled with the farm flow conditions, especially the inflow wind direction. Therefore, wind farm flow control solutions that aim to regulate wind farm performance need to consider wind direction variability to be effective (Starke et al., 2021).

First of all, the use of robust control solutions that account for the uncertainties in input wind direction signals in their
calculations have been shown to alleviate some of the problems associated with wind direction variability and bring about improvements in wake steering control (Rott et al., 2018; Quick et al., 2020) (Section 7.1). More understanding of the uncertainty bounds on control system inputs are needed in order to better evaluate the benefits and limitations of any given control approach (Shapiro et al., 2022).




Secondly, accurate wind direction measurement and estimation are critical for the implementation of successful wind farm
and turbine controllers. Collective yaw control has been shown to offer slight improvements in power productions alongside
substantial reductions in yaw activation (Bossanyi, 2019; Van Der Hoek et al., 2021) (Section 7.2). However, benefits were
only seen in simple simulated scenarios over short time intervals, therefore more investigation is necessary.

## 8  Discussion

Wind farms are routinely subjected to changing wind directions, yet the effect on wind farms under realistic wind direction
changes remains understudied (Shapiro et al., 2022). Accounting for the dynamic effect of these changes in high fidelity wind
farm flow models has been shown to improve power output estimates (Munters et al., 2016) and result in more effective yaw
and wake steering controllers compared to approaches that assume a static wind direction (Rott et al., 2018; Simley et al.,
2020).

Testing and validation of new control systems in simulations is essential before deployment in real world wind farms and
relies on the use of wind farm flow models. These models need to make simplifying assumptions about the full flow field, and
neglect most or at least some of the variability present in real-world conditions. These necessary assumptions have led to wind
direction variability being mostly overlooked when it comes to assesing overall wind farm performance.

As discussed, most of the control-oriented modelling of wind direction up to the present has only been analysed over
short time periods, in limited atmospheric conditions and with a focus purely on the objective of power gain and not overall
performance improvements. Ultimately, research needs to assess the true impact of wind direction on wind farm performance,
specifically the impact on LCOE. Hence, Section 8.1 introduces the critical challenges to be solved for this objective to be
achieved.

### 8.1  Critical Challenges

From the literature, three critical technical challenges in control-oriented wind direction research can be identified. The three
challenges are,

1. **Improved measurement of wind direction** - Reliable and comprehensive wind direction data needs to be obtained
   for model testing and validation along with agreement on standards of how wind direction should be measured and
   conditioned before use, particularly in relation to flow distortions, atmospheric stability and height above the surface.
   Measurement campaigns to produce large data sets for this specific purpose are imperative.

2. **Modelling realistic wind direction spatial and temporal variability with reasonable accuracy and computational
   cost** - Creation of validated and tested statistical and/or physical models that cover the full envelope of operational con-
   ditions are necessary to perform less computationally intensive data-driven wind farm flow simulations. Complementary
   to this, there is a parallel need for continued development of high fidelity meso-scale coupled LES models to analyse





the important physical drivers of variability in more detail, as well as to better understand the interactions between wind direction variability and wind turbine wakes.

3. **Development of a detailed scientific understanding of performance effects of wind direction variability and yaw misalignment on wind turbines and wind farms** - Extensive measurement campaigns are required to record turbine loads and power production data coupled with wind direction and yaw misalignment data. First and foremost, these measurements would allow for a proper scientific understanding of cause and effect. Only then can better control-oriented models be designed and evaluated for prediction of power production and loads under yaw misalignment, which in turn can inform controller synthesis.

Addressing these challenges requires interdisciplinary research efforts that combine expertise from meteorology, control engineering, data science, and wind energy systems. Whilst the three critical challenges outlined above must be accomplished, there is a further critical dissemination challenge of embedding wind direction models within turbine and farm flow control research, with respect to both design and testing. The first steps in this process are,

- Knowledge exchange and guidance for researchers in adjacent research areas as to the importance of wind direction modelling.

- Making wind direction models freely available and usable by researchers within other areas.

Tackling these challenges will have an important positive impact on wind turbine and farm modelling, design, and operational analysis. It will contribute to improving performance and reliability, and ultimately help to reduce the LCOE of wind energy.

## 9 Conclusions

Wind direction variability plays a critical role in the operation and performance of wind farms. It is inherently non-linear and non-stationary due to complex atmospheric processes and the turbulent nature of wind flows. Additionally, wind direction varies both spatially and temporally, making it challenging to develop models that capture all of these effects at once. Site specific conditions, such as wake and terrain effects, can also play a substantial role in wind farm performance.

The direction of the inflow relative to the rotor plane affects the aerodynamics of wind turbines in complex and unclear ways, which has implication for overall performance in terms of both power and loading. Incorporating such effects into wind farm flow models is important for controller design and testing. Wind farms are routinely subjected to changing wind directions, sometimes extreme changes, that need to be taken into account in wind farm flow control solutions that aim to regulate wind farm performance (Starke et al., 2021). However, the uncertainty in wind direction measurements makes the assessment and implementation of control solutions more challenging, since accurate representations of cause and effect relationships for control purposes is difficult. The challenge is compounded by the fact that the behaviour of wind turbines and wind farm flow under realistic wind direction changes remains understudied (Shapiro et al., 2022).





The design of the yaw control system needs to incorporate important aspects of both physical analysis and statistical analysis, such that it can optimise the turbine's operation while minimising the LCOE. The critical challenges associated with achieving this optimisation can be separated into three broad categories. These are; **improved measurements of wind direction**, **realistic dynamic wind direction modelling** and **farm and turbine performance effects of wind direction variability yaw misalignment**.

As wind energy plays an increasingly important role in global energy production, the development of accurate and versatile control-oriented models will ensure the continued performance, reliability, efficiency and competitiveness of wind energy in the years to come.

*Author contributions.* Conceptualisation, S.D., A.S., and E.H.; writing—original draft preparation, S.D.; writing—review and editing, S.D., A.S. and E.H.; visualisation, S.D.; supervision, A.S., and E.H.; All authors have read and agreed to the published version of the manuscript.

*Competing interests.* The contact author has declared that none of the authors have any competing interests.

*Acknowledgements.* SD is funded by EP/S023801/1 EPSRC Centre for Doctoral Training in Wind and Marine Energy Systems and Structures, and would like to thank all of the staff involved in the CDT. EH is funded by a Brunel Fellowship from the Royal Commission for the Exhibition of 1851.



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
