# Peer review of "Control-Oriented Modelling of Wind Direction Variability"

_Wind Energy Science, 2023_

## Author Comment (AC2)

**Authors' Response to Reviewer 1**

> **General Comments.** I have several comments to help clarify some points throughout the text (considering the increased length of a review paper). Regardless, I think this manuscript is a comprehensive review of its stated objectives and should be accepted for publication.

**Response:** Thank you very much for your feedback.

Allow us a moment to express our gratitude for the time and effort you invested in reviewing the paper. Your comments are insightful and constructive. They are extremely valuable in improving the quality and clarity of our work.

**Comment 1**

Line 73/103: There are several points throughout Section 2 where descriptions like "long term," "small scale," "low frequency," etc., are used to describe certain regimes of physical processes. Section 2.1 ties the definition of some of these descriptors to specific length and time scales, but I'm not sure if those definitions are consistent throughout Section 2. I think it could be helpful to specify what these descriptions of scale mean when they come up, or clearly summarize them at the beginning.

**Response:**

Section 2 is meant to paint a picture of the relevant phenomena that need to be considered and therefore broad categories are introduced. We agree that more precise language is needed and words like "long term," "small scale," "low frequency," are quite vague, especially when not in the context of micro, meso and synoptic scale, introduced at the start of section 2.1. We will review this section in the manuscript and adjust accordingly.

**Comment 2**

Line 121: This clockwise rotation with height is only true in the Northern Hemisphere.

**Response:**

Thank you for pointing this out, it is a simple edit which we will make to the manuscript.

**Comment 3**

Line 132: This sentence suggests that time scales of 10 minutes are significant for wind farm flow control. Can that be explicitly stated/summarized somewhere? The authors argue that a range of length and time scales are important for modeling wind direction variability, but the control problem itself is rooted in a specific range of length and time scales. I think that added context could be helpful to the reader.

**Response:**

We had a discussion about this and more context is needed here. It is an implication of the control architecture and yaw rates of modern wind turbines (in the order of minutes), therefore direction changes with enough magnitude in the order of minutes could elicit a control response. The wording needs to be changed here to convey that idea properly. We will update the manuscript with these changes.

**Comment 4**

Line 161: What is meant by "current developments in high-fidelity farm flow models" that are out of reach to the majority of the research community? Are the authors referring to direct numerical simulations? And then the argument is that large eddy simulations are more practical and within reach? I think this wording is slightly confusing because "high-fidelity" is linked to large eddy simulations at Line 171.

**Response:**

The sentence refers to LES with concurrent precursor simulations which are used to model dynamic wind direction changes in the wind farm LES which are less accessible than LES with preset initial and boundary conditions. We will change the manuscript to emphasise that here and make it clear and consistent with the rest of the paper.

**Comment 5**

Eq. 1: Some parentheses could be added to clarify order of operations in this equation.

**Response:**

Thank you for highlighting this, we will edit the equation in the manuscript to fix this. We also plan to add the calculation of the signed minimum angular distance to complement this equation.

**Comment 6**

Fig. 2: This figure needs labels and more context to provide value to the reader; I personally didn't understand much from these vector diagrams. Related: how is the vector defined in the complex plane (Line 290)?

**Response:**

The idea was to make the diagram as minimal as possible but we definitely lost interpretability along the way. We will review this figure and add more text to it. As for your other question, in the complex plane, the wind direction angles are cast as the argument of complex numbers in exponential form with unit radius, which makes calculation of the mean angle a little more intuitive. If you imagine the vector summation of the complex numbers and then measuring the angle of the resultant vector as in the figure, you arrive at the circular mean, which is demonstrated in the figure. That connection needs to be more explicit and we will edit this part in the manuscript.
* * *
**Comment 7**

Line 346: Yaw misalignment is mentioned before this point, but I'm not sure if it is clarified that is in reference to the hub height time-averaged wind direction. There is certainly instantaneous misalignment at any time, but the focus is on error relative to time-averaged (over the relevant time scale of the control problem) wind direction.
* * *
**Response:**

Yaw misalignment is not well defined in general. We take it to mean any (instantaneous or time-averaged, static or dynamic) difference between the rotor axis angle and the hub height wind direction. However, as you point out, the instantaneous yaw misalignment is not useful for our purposes by itself and we are much more interested in the time-averaged misalignment (in the order of minutes). We will review the wording to make sure it is clear what we are talking about at all times. We will also make sure other related terms like yaw error and yaw offset are consistent and well defined.

**Comment 8**

Line 393: Is there a reason why the Heck, Johlas, and Howland (2023) citation is not used here in addition to the Howland et al. (2020) citation? The 2023 paper also derives a physics-based model for the power ratio/power reduction factor from first principles, assuming that the thrust force characteristics as a function of yaw misalignment are known.

**Response:**

We noted the Heck, Johlas, and Howland (2023) paper while writing the review but must have overlooked it when writing this section (perhaps it was published after this section was completed). We will add this paper to the manuscript.

**Comment 9**

Line 421: Rotating the rotor to the "right" is unclear - does this refer to clockwise rotation?

**Response:**

We agree that rotating to the right is not clear and depends on the readers perspective. From a top down view of the turbine we meant clockwise rotation. We will change the wording here to reflect that.

**Comment 10**

Line 506: Why are the yaw signals "mostly wrong?" Based on the description of the Draxl paper in the next sentence, I would think it's more appropriate to say that the signal is wrong at start-up?

**Response:**

Draxl wrote about the wind direction measurements at Horns Rev being mostly wrong, but only when the turbines are not operating, which is different from what we have said here. We will change this part in the manuscript to make it accurate.

> **Comment 11**
>
> Line 564: Could slightly more description be given for these measurement-free yaw control methods? Based on the text here, I can't wrap my head around how these techniques could work.

**Response:**

The general idea is to infer the wind direction signal from other more reliable measurements. There are only a couple of examples of this in the literature and so it's not given much space in the text. It is skipped over somewhat to focus more on the other techniques. However, you're right that it warrants more explanation since these techniques are not widely known and might be of interest to a lot of people.

---

## Author Comment (AC3)

**Authors' Response to Reviewer 2**

> **General Comments.** Overall, the manuscript gives a very good overview of many research questions and aspects and clearly shows that a good understanding of wind direction variability is crucial in many fields of wind energy application and that this understanding is still incomplete. This also shows the need for further research and provides a good introduction to the topic, especially for new scientists. Overall, I therefore consider this paper to be an important contribution.

**Response:** Thank you very much for your feedback.

We appreciate the attention you paid to our paper and the insightful suggestions for improvement. Your expertise in the subject area is clear, and we are grateful to have benefited from your knowledge.

**Comment 1**

line 25: "EC 61400-1" Here, the "I" for IEC is missing.

**Response:**

Thank you, this error will be corrected.

**Comment 2**

line 59: "as as" is a typo.

**Response:**

Thank you, this error will be corrected.

**Comment 3**

line 206: "the maximum horizontal resolution of these models is too large to allow them to accurately investigate intra-wind farm effects"; I assume you mean that the resolution of a mesoscale model like WRF is not high enough to resolve intra-wind farm effects, is that correct?

**Response:**

Yes, the current maximum resolution of mesoscale models like WRF is in the order of kilometres in space and in the order of minutes in time so are not useful for resolving the flow around turbines. They are useful in studying general wind farm flow effects, for example inter-wind farm wakes and the development of wind farm boundary layers. We decided to exclude discussion on that topic since it strayed a little far away from the control-oriented part, but a paragraph or so in the revised manuscript would make the point more explicit here.

> ## Comment 4
>
> line 285 eq(1). The definition for the absolute minimum distance according to (Farrugia2009) is fine, although I agree with RC1 that additional parentheses can be helpful. As stated in the author's comment, this will be updated.
>
> However, the equation given in (Farrugia2009) is not complete, in my opinion. (Note that Farrugia substracts the second variable from the first, which is the opposite of eq1 of this paper). E.g. (in degrees for simplicity): $\Delta(190°, 0°)$. The result should be $-170°$. This case is not covered in Eq1 of Farrugia2009 in the "if cases".
>
> To simplify things, I suggest the shift trick from Rott2018: $\Delta(A, B) = ((B - A + \pi) \mod 2\pi) - \pi$
>
> The only difference is that with the shift trick per definition if the distance of A and B is precisely $\pi$, the result is $\Delta(A, B) = -\pi$, which is equivalent to a rotation of $\pi$, but in opposition to the definition of the sign in Farrugia2009. But this allows us to define the $\Delta$ operator directly, without if cases. And from this, the absolute can be defined, and not vice versa.

**Response:**

There seems to be a few different ways of calculating this signed angular distance, we also arrived at one independently of both Farrugia2009 and Rott2018. We must have missed the definition from Rott2018 but it appears to be a more succinct and probably computationally faster calculation compared to both our approach and Farrugia2009. We will include this in the revised manuscript with a short discussion about each approach.

**Comment 5**

line 298: Eq3 calculates the "regular" variance (or variance "on the line", which is the term that Fisher uses), which is the expected quadratic difference of the individual samples to the "arithmetic" mean. The circular variance is defined in Fisher1995 (Statistical analysis of circular data) and also in (Cremers and Klugkist, 2018) (`https://www.frontiersin.org/articles/10.3389/fpsyg.2018.02040/full`) as $v = 1 - \overline{R}$, where $\overline{R}$ is the mean resultant length (dashed line in Figure2 divided by n). The circular variance is a measure of the variability of the data, like the variance "on the line", but it is bounded between 0 and 1 and is mathematically different from the variance "on the line". Calculating the circular standard deviation is also more complicated than the standard deviation "on the line". The problem with the "on the line" statistics arises because distributions on the circle wrap around; therefore, the interpretation of the statistics is not the same (see Note 2 in Fisher1995).

However, for the application of short-term wind data, the widths of wind direction distributions are comparatively very small, and the chances of wrapping around are basically 0. Therefore, it makes sense to use the common terminology like variance and standard deviation, although this is mathematically not 100 % correct. In this review paper, however, the difference should be emphasised very clearly.

**Response:**

Thank you for pointing this out, we suspected this section needed more careful consideration. While researching the paper, we came across both the "on the line" description and the bounded circular definition. Only the "on the line" method was included for brevity. However, it is clear that there is a deeper discussion to be had here around the points you raised and the various ways the mean and variance of circular variables can be defined. We will address these points in the revised manuscript to give a complete picture.

**Comment 6**

line 589: "Wind Direction Forecasting" is an essential topic for many applications that significantly differ in time scales. Long-term forecasts, short-term forecasts, and shortest-term forecasts (nowcasts). Since this is a paper about Control-oriented modelling, it should be clear that the forecasts are in the shortest-term range, but the reader could be reminded at the beginning of this section to avoid confusion.

**Response:**

Agreed, there are different time-scales over which you might want to forecast the wind direction which correspond to various applications. We briefly introduce forecasting in the bullet points of section 6, however it is a short sentence and it is not clear why we are focusing only on the short-term range. We will correct this in the revised manuscript.

---

## Author Response (AR1)

**Authors' Response to Reviewer 1**

> **General Comments.** I have several comments to help clarify some points throughout the text (considering the increased length of a review paper). Regardless, I think this manuscript is a comprehensive review of its stated objectives and should be accepted for publication.

**Response:** Thank you very much for your feedback.

Allow us a moment to express our gratitude for the time and effort you invested in reviewing the paper. Your comments are insightful and constructive. They are extremely valuable in improving the quality and clarity of our work. **Note that the line numbers of edits correspond to the line numbers in the revised manuscript.**

**Comment 1**

Line 73/103: There are several points throughout Section 2 where descriptions like "long term," "small scale," "low frequency," etc., are used to describe certain regimes of physical processes. Section 2.1 ties the definition of some of these descriptors to specific length and time scales, but I'm not sure if those definitions are consistent throughout Section 2. I think it could be helpful to specify what these descriptions of scale mean when they come up, or clearly summarize them at the beginning.

**Response:**

Thank you for pointing this out, section 2 is meant to paint a picture of the relevant phenomena that need to be considered and therefore broad categories are introduced. We agree that more precise language is needed and phrases like "long term," "small scale," "low frequency," are quite vague, especially when not in the context of micro, meso and synoptic scale, introduced at the start of section 2.1.

We have reviewed Section 2 and changed those phrases throughout to either microscale, mesoscale or synoptic scale to match with Figure 1 and the terminology introduced in Section 2.1. "Low frequency" has been kept because it is used in the context of mesoscale processes and timescale of more than 10 minutes therefore we think the meaning is clearer here.
* * *
**Comment 2**

Line 121: This clockwise rotation with height is only true in the Northern Hemisphere.

**Response:**

Thank you for noticing this error, we have edited this out of the manuscript so it simply reads "rotation with height" (Line 125).
* * *
**Comment 3**

Line 132: This sentence suggests that time scales of 10 minutes are significant for wind farm flow control. Can that be explicitly stated/summarized somewhere? The authors argue that a range of length and time scales are important for modeling wind direction variability, but the control problem itself is rooted in a specific range of length and time scales. I think that added context could be helpful to the reader.

**Response:**

Thank you. It is an implication of the control architecture and yaw rates of modern wind turbines (in the order of minutes), therefore direction changes with enough magnitude in the order of minutes could elicit a control response.

We have changed the wording and included a reference to Section 5 where the details of conventional yaw systems are discussed, like hysteresis dead-bands and yaw rates, concepts which are helpful in understanding the relevant time scales. (Lines 133 to 136)

> ## Comment 4
> Line 161: What is meant by "current developments in high-fidelity farm flow models" that are out of reach to the majority of the research community? Are the authors referring to direct numerical simulations? And then the argument is that large eddy simulations are more practical and within reach? I think this wording is slightly confusing because "high-fidelity" is linked to large eddy simulations at Line 171.

**Response:**

This sentence refers to LES with concurrent precursor simulations which are used to model dynamic wind direction changes in wind farm LES which are less accessible than LES with preset initial and boundary conditions but more accessible than DNS. We agree that it is unclear that this is what we mean here.

Thank you, we have updated the manuscript to emphasise our meaning here (introducing LES with precursor simulations) and make it clear and consistent with the other parts of the paper. (Lines 165 and 176).

> ## Comment 5
> Eq. 1: Some parentheses could be added to clarify order of operations in this equation.

**Response:**

Thank you for highlighting this, we have edited the equation in the manuscript to fix this. We have also added the calculation of the signed minimum angular distance to complement this equation. (Eq. 3 and Eq. 4)

**Comment 6**

Fig. 2: This figure needs labels and more context to provide value to the reader; I personally didn't understand much from these vector diagrams. Related: how is the vector defined in the complex plane (Line 290)?

**Response:**

The idea was to make the diagram as minimal as possible but we definitely lost interpretability along the way.

Thank you, we have now updated this figure and added more text to it. The connection with the representation in the complex plane has been made more explicit by adding labels and more detail to the text and the figure caption as well. (Fig. 2 and lines 287 to 291)

**Comment 7**

Line 346: Yaw misalignment is mentioned before this point, but I'm not sure if it is clarified that is in reference to the hub height time-averaged wind direction. There is certainly instantaneous misalignment at any time, but the focus is on error relative to time-averaged (over the relevant time scale of the control problem) wind direction.

**Response:**

Thank you, this a good point that we had not highlighted. However, we would also point out that there are also instances (e.g. controller input) where instantaneous $\theta_n acalle$

values are used with instantaneous $\theta_w ind$ values, as well as cases where both thetas as time averages. We capture each of these cases in the revised wording when defining $\gamma_E$.

**Comment 8**

Line 393: Is there a reason why the Heck, Johlas, and Howland (2023) citation is not used here in addition to the Howland et al. (2020) citation? The 2023 paper also derives a physics-based model for the power ratio/power reduction factor from first principles, assuming that the thrust force characteristics as a function of yaw misalignment are known.

**Response:**

We noted the Heck, Johlas, and Howland (2023) paper while writing the review but must have overlooked it when writing this section (perhaps it was published after this section was completed).

Thank you for highlighting it, we have added a short discussion of the paper to the manuscript and linked it with the previous study. (Line 420)

**Comment 9**

Line 421: Rotating the rotor to the "right" is unclear - does this refer to clockwise rotation?

**Response:**

We agree that rotating to the right is not clear and depends on the readers perspective. From a top down view of the turbine we meant clockwise rotation. Therefore, we have changed the wording here to "clockwise rotation". Thank you. (Line 446)

**Comment 10**

Line 506: Why are the yaw signals "mostly wrong?" Based on the description of the Draxl paper in the next sentence, I would think it's more appropriate to say that the signal is wrong at start-up?

**Response:**

Draxl wrote about the wind direction measurements at Horns Rev being mostly wrong, but only when the turbines are not operating, which is different from what we have said here.

Thanks for noticing this, we have updated this part in the manuscript to reflect that. (Lines 530 to 533)

**Comment 11**

Line 564: Could slightly more description be given for these measurement-free yaw control methods? Based on the text here, I can't wrap my head around how these techniques could work.

**Response:**

The general idea is to infer the wind direction signal from other more reliable measurements. There are only a few examples of this in the literature and so it's not given much space in the text. It is skipped over somewhat to focus more on the other techniques.

However, thank you, you're right that it warrants more explanation since these techniques are not widely known and might be of interest to a lot of people. Therefore, we have added a short description of how they work based on the two most recent papers cited (Tsioumas et al., 2017 and Karami et al., 2021). (Lines 590 to 594)

**Authors' Response to Reviewer 2**

> **General Comments.** Overall, the manuscript gives a very good overview of many research questions and aspects and clearly shows that a good understanding of wind direction variability is crucial in many fields of wind energy application and that this understanding is still incomplete. This also shows the need for further research and provides a good introduction to the topic, especially for new scientists. Overall, I therefore consider this paper to be an important contribution.

**Response:** Thank you very much for your feedback.

We appreciate the attention you paid to our paper and the insightful suggestions for improvement. Your expertise in the subject area is clear, and we are grateful to have benefited from your knowledge. **Note that the line numbers of edits correspond to the line numbers in the revised manuscript.**

**Comment 1**

line 25: "EC 61400-1" Here, the "I" for IEC is missing.

**Response:**

Thank you, this error has been corrected. (Line 25)

**Comment 2**

line 59: "as as" is a typo.

**Response:**

Thank you for pointing this out, it has now been corrected. (Line 59)

line 206: "the maximum horizontal resolution of these models is too large to allow them to accurately investigate intra-wind farm effects"; I assume you mean that the resolution of a mesoscale model like WRF is not high enough to resolve intra-wind farm effects, is that correct?

**Response:**

Yes, the current maximum resolution of mesoscale models like WRF is in the order of kilometres in space and in the order of minutes in time so are not useful for resolving the flow around turbines. They are useful in studying general wind farm flow effects, for example inter-wind farm wakes and the development of wind farm boundary layers.

We decided to exclude discussion on that topic since it strayed a little far away from the control-oriented part, but thank you for raising the point. It is an important topic so we have added a sentence to the revised manuscript which introduces the context in which they can be useful. (Lines 213 to 214)

> ### Comment 4
>
> line 285 eq(1). The definition for the absolute minimum distance according to (Farrugia2009) is fine, although I agree with RC1 that additional parentheses can be helpful. As stated in the author's comment, this will be updated.
>
> However, the equation given in (Farrugia2009) is not complete, in my opinion. (Note that Farrugia substracts the second variable from the first, which is the opposite of eq1 of this paper). E.g. (in degrees for simplicity): $\Delta(190°, 0°)$. The result should be $-170°$. This case is not covered in Eq1 of Farrugia2009 in the "if cases".
>
> To simplify things, I suggest the shift trick from Rott2018: $\Delta(A, B) = ((B - A + \pi) \mod 2\pi) - \pi$
>
> The only difference is that with the shift trick per definition if the distance of A and B is precisely $\pi$, the result is $\Delta(A, B) = -\pi$, which is equivalent to a rotation of $\pi$, but in opposition to the definition of the sign in Farrugia2009. But this allows us to define the $\Delta$ operator directly, without if cases. And from this, the absolute can be defined, and not vice versa.

**Response:**

Thank you for your detailed feedback. We have included the definition from Rott2018 and have a short sentence on each approach. We haven't discussed each approach in any great detail but agree that the simplest and the only complete approach is by Rott2018 which we have highlighted. (Eq. 3 and Eq. 4, lines 304 to 313)

**Comment 5**

line 298: Eq3 calculates the "regular" variance (or variance "on the line", which is the term that Fisher uses), which is the expected quadratic difference of the individual samples to the "arithmetic" mean. The circular variance is defined in Fisher1995 (Statistical analysis of circular data) and also in (Cremers and Klugkist, 2018) (`https://www.frontiersin.org/articles/10.3389/fpsyg.2018.02040/full`) as $v = 1 - \overline{R}$, where $\overline{R}$ is the mean resultant length (dashed line in Figure2 divided by n). The circular variance is a measure of the variability of the data, like the variance "on the line", but it is bounded between 0 and 1 and is mathematically different from the variance "on the line". Calculating the circular standard deviation is also more complicated than the standard deviation "on the line". The problem with the "on the line" statistics arises because distributions on the circle wrap around; therefore, the interpretation of the statistics is not the same (see Note 2 in Fisher1995).

However, for the application of short-term wind data, the widths of wind direction distributions are comparatively very small, and the chances of wrapping around are basically 0. Therefore, it makes sense to use the common terminology like variance and standard deviation, although this is mathematically not 100 % correct. In this review paper, however, the difference should be emphasised very clearly.

**Response:**

Thank you for pointing this out, we suspected this section needed more careful consideration. While researching the paper, we came across both the "on the line" description and the bounded circular definition. Only the "on the line" method was included for brevity.

We have now included both approaches and followed your suggestion to recommend the on the line approach when the range of data is narrow enough to not cause any complications. We have added a footnote to explain when the linear approach becomes invalid. (Lines 300 to 303)

**Comment 6**

line 589: "Wind Direction Forecasting" is an essential topic for many applications that significantly differ in time scales. Long-term forecasts, short-term forecasts, and shortest-term forecasts (nowcasts). Since this is a paper about Control-oriented modelling, it should be clear that the forecasts are in the shortest-term range, but the reader could be reminded at the beginning of this section to avoid confusion.

**Response:**

Thank you, we agree there are different time-scales over which you might want to forecast the wind direction which correspond to various applications. We briefly introduce forecasting in the bullet points of section 6, however it is a short sentence and it is not clear what time horizon we are focusing on.

We have now introduced the term very short term (in the order of minutes) to correspond with the literature. (Line 603 and Section 6.1)

---

## Referee Report (RR1)

**Review WES-2023-93**

The presented review article is very comprehensive and covers in detail all important areas of wind direction variability related to wind turbines and their control. It summarizes important results on wind direction variability from a large number of publications and provides a comprehensive basis for current and future research.

There are two minor comments I would like to make regarding this manuscript:

1. Equation (1) implies that $z_R$ is the mean vector of unit vectors $z_i$. Therefore, its length/norm is bounded by $0 \leq |z_R| \leq 1$. In Figure 2, $z_R$ is shown as the sum of the unit vectors $z_i$. For $\arg(z_R)$, this makes no difference since the angle is the same, but the length of the vector is used to define the circular variance $v_R$ in Equation (2). Either the caption of the figure should clarify that $z_R$ in this figure illustrates the sum and not the mean (so it is 3 times as long), or the dashed arrow should be adjusted to its correct length.

2. In Equation (5), the formula for the linear variance is given, which uses the distance to the mean $\bar{\theta}$. It is important to clarify that the mean $\bar{\theta}$ being referred to in this context is the linear mean. This distinction is important because using the circular mean in Equation (5) would lead to a different result and not resemble the linear standard deviation.

---

## Author Response (AR2)

**Authors' Response to Reviewer 2**

> **General Comments.** The presented review article is very comprehensive and covers in detail all important areas of wind direction variability related to wind turbines and their control. It summarizes important results on wind direction variability from a large number of publications and provides a comprehensive basis for current and future research.

**Response:** Thank you very much for your feedback.

Your feedback has fundamentally improved the quality of the review paper and we are extremely grateful for it.

> ## Comment 1
>
> Equation (1) implies that $z_R$ is the mean vector of unit vectors $z_i$. Therefore, its length/norm is bounded by $0 \leq |z_R| \leq 1$. In Figure 2, $z_R$ is shown as the sum of the unit vectors $z_i$. For $\arg(z_R)$, this makes no difference since the angle is the same, but the length of the vector is used to define the circular variance $v_R$ in Equation (2). Either the caption of the figure should clarify that $z_R$ in this figure illustrates the sum and not the mean (so it is 3 times as long), or the dashed arrow should be adjusted to its correct length.

**Response:**

Thank you, this error has been corrected by changing $z_R$ in Equation 1 to $z_R/N$ and adding the subscripts $C$ and $L$ to make it clearer when circular or linear statistics are being referenced. Now, $v_R$ is referred to as $v_C$, the circular mean as $\bar{\theta}_C$, and the linear mean as $\bar{\theta}_L$.

**Comment 2**

In Equation (5), the formula for the linear variance is given, which uses the distance to the mean $\bar{\theta}$. It is important to clarify that the mean $\bar{\theta}$ being referred to in this context is the linear mean. This distinction is important because using the circular mean in Equation (5) would lead to a different result and not resemble the linear standard deviation.

**Response:**

Thank you, we've added the subscripts $C$ and $L$ to make it clearer when circular or linear statistics are being referenced. Now, it is clear which calculation of the mean value we are referencing.